# CvkR is a MerR-type transcriptional repressor of class 2 type V-K CRISPR-associated transposase systems

Marcus Ziemann [1,7], Viktoria Reimann [1,7], Yajing Liang [2,3,4,7], Yue Shi [2,3,4], Honglei Ma [2,3,4,5], Yuman Xie [2,5], Hui Li [2,5], Tao Zhu [2,3,4,5,8] ✉, Xuefeng Lu [2,3,4,5,6,8] ✉ & Wolfgang R. Hess [1,8] ✉

Certain CRISPR-Cas elements integrate into Tn7-like transposons, forming CRISPR-associated transposon (CAST) systems. How the activity of these systems is controlled in situ has remained largely unknown. Here we characterize the MerR-type transcriptional regulator Alr3614 that is encoded by one of the CAST (AnCAST) system genes in the genome of cyanobacterium *Anabaena* sp. PCC 7120. We identify a number of Alr3614 homologs across cyanobacteria and suggest naming these regulators CvkR for Cas V-K repressors. Alr3614/CvkR is translated from leaderless mRNA and represses the AnCAST core modules *cas12k* and *tnsB* directly, and indirectly the abundance of the tracrCRISPR RNA. We identify a widely conserved CvkR binding motif 5′-AnnA-CATnATGTnnT-3′. Crystal structure of CvkR at 1.6 Å resolution reveals that it comprises distinct dimerization and potential effector-binding domains and that it assembles into a homodimer, representing a discrete structural subfamily of MerR regulators. CvkR repressors are at the core of a widely conserved regulatory mechanism that controls type V-K CAST systems.

Native Clustered Regularly Interspaced Short Palindromic Repeats (CRISPRs) and CRISPR-associated (Cas) proteins are well characterized for their function as RNA-based adaptive and inheritable immune systems present in many bacteria and archaea[1–6]. Multiple genetic approaches developed from these native CRISPR-Cas systems have become popular for the manipulation of gene expression and genome editing[7–9].

CRISPR-Cas systems are highly diverse and are classified into 2 classes, 6 types and 33 subtypes[10]. Recently, a remarkable group of derivatives has been discovered that constitute hybrids of Tn7-like transposons and CRISPR systems encoding Cas12k effectors with naturally inactivated nuclease domains[11] or encoding Cascade

complexes lacking the Cas3 nuclease component[12,13]. The respective transposon-associated CRISPR systems include class 1 type I-F, I-B, and class 2 type V-K systems[13,14]. These systems, called CRISPR-associated transposons (CASTs), are capable of catalyzing the transposition of mobile genetic elements guided by crRNAs, while the type I-B associated systems use a dedicated TniQ/TnsD protein-based homing mechanism[13].

The system characterized from *Vibrio cholerae* consists of genes encoding the subtype I-F CRISPR-Cas proteins Cas6, Cas7, a Cas5-Cas8 fusion protein and genes encoding the transposon proteins TnsA, TnsB, TnsC, and TniQ[12,14]. A single instance of a class 1 type I-B and of several class 2 type V-K CAST systems have been reported in several

[1]Faculty of Biology, Institute of Biology III, Genetics and Experimental Bioinformatics, University of Freiburg, Schänzlestr. 1, Freiburg D-79104, Germany. [2]Qingdao Institute of Bioenergy and Bioprocess Technology (QIBEBT), Chinese Academy of Sciences, No.189 Songling Road, Qingdao 266101, China. [3]Shandong Energy Institute, Qingdao 266101, China. [4]Qingdao New Energy Shandong Laboratory, Qingdao 266101, China. [5]University of Chinese Academy of Sciences, Beijing 100049, China. [6]Laboratory for Marine Biology and Biotechnology, Qingdao National Laboratory for Marine Science and Technology, Qingdao 266237, China. [7]These authors contributed equally: Marcus Ziemann, Viktoria Reimann, Yajing Liang. [8]These authors jointly supervised this work: Tao Zhu, Xuefeng Lu, Wolfgang R. Hess. ✉e-mail: zhutao@qibebt.ac.cn; lvxf@qibebt.ac.cn; wolfgang.hess@biologie.uni-freiburg.de

different cyanobacteria, which constitute a rich natural resource for these systems[11,13,15].

The V-K CAST systems, first characterized in *Scytonema hofmanni*[11], contain genes encoding the effector complex subunit Cas12k and the Tn7-like transposase subunits TnsB, TnsC and TniQ, while *tnsA* is lacking. A surprising finding has been that the ribosomal protein S15 is recruited to the target DNA-bound Cas12k-transposon complex[16]. Targeting transposition by these CAST systems depends on the DNA-crRNA interaction facilitated by the effector protein Cas12k[11]. The TnsC transposon then forms helical polymers around the DNA supported by ATP binding[17]. The growth in the 5′ to 3′ direction is stopped by TniQ binding at the filament end concomitantly connecting the TnsC filament with Cas12k. On the other filament end, the Mu-like transposase TnsB then starts to integrate the transposon[17]. In addition to the genes encoding transposase and effector proteins, all of these systems contain various numbers of cargo genes. Novel genetic approaches have been developed from the different Tn7-CRISPR-Cas hybrid systems[18–20], underlining that the better characterization of such systems is of both fundamental and applied interest.

While the paradigm is that native CRISPR-Cas systems primarily protect genome integrity against mobile genetic elements, the CAST systems seem to violate this paradigm since they constitute transposable elements by definition. Because of these contradicting functions, the tight regulation of these systems can be expected. Indeed, CAST systems have also been reported to contain a gene encoding a putative MerR-type transcriptional regulator[15]. The classical MerR-type regulators uniquely modulate gene expression by binding the core promoter region with atypical long spacers between the −35 and −10 elements[21–23]. However, the association of MerR regulators with CAST systems has not been systematically investigated, nor has its function been addressed experimentally thus far.

We have studied the CRISPR-Cas systems in *Anabaena* (*Nostoc*) sp. PCC 7120 (from here: *Anabaena* 7120), a multicellular nitrogen-fixing model cyanobacterium with a CRISPR-rich chromosome of eleven CRISPR-like repeat-spacer cassettes. All of them are transcribed[15], and based on the specificities of the cognate Cas6 maturation endonucleases, five of these arrays were assigned to a type III-D and another five to a type I-D CRISPR-Cas system[24], while the remaining array (CR_9) belongs to a separate CRISPR type with all the hallmarks of a CAST system[15,24].

Here, we first scrutinize the association between putative transcriptional regulators and cyanobacterial CAST systems and find these to belong to three different types, with the majority being MerR-like regulators. We then investigate the MerR-like Alr3614 transcriptional regulator belonging to the *Anabaena* 7120 CRISPR-associated transposase (AnCAST) system. We establish that both the *cas12k* gene *all3613* and the *merR*-like gene *alr3614* are translated from leaderless

mRNAs and that Alr3614 functions as a repressor of the AnCAST system. We identify a conserved 15 nt long Alr3614 binding motif by DNase I footprinting and gel shift assays as well as bioinformatic analysis. Crystal structure analysis reveal that Alr3614 adopts a unique dimerization pattern and packing mode in the C-terminal sensor region, while it lacks a recognizable effector-binding site compared to other MerR family members. Thus, Alr3614 is an unusual MerR family regulator from a structural perspective. We suggest naming Alr3614 and its homologs Cas V-K repressor (CvkR) encoded by the gene *cvkR*.

## Results

### Architecture of cyanobacterial CAST systems

Starting from the known CAST components, we searched for conserved genes and genetic elements in their vicinity. These elements included the left and right ends of the transposon, the neighboring tRNA, CRISPR arrays and tracrRNA, the transposase genes (*tnsB*, *tnsC* and *tniQ*), and genes in reverse orientation next to the start codon of *cas12k* predicted to encode small DNA-binding proteins. We identified 118 CAST systems with a clear *cas12k* gene in 88 different strains. The majority of these were found in the Nostocales (60%), Chroococcales (15%), and Pseudanabaenales (11%), complemented by a small number of CAST systems in the Oscillatoriales, Pleurocapsales, Spirulinales and Synechococcales cyanobacteria (Supplementary Data 1). Three additional CAST systems were identified in unclassified filamentous cyanobacteria (CCT1, CCP2, and 4).

From this analysis, we delineated the general structure of this type of CAST system (Fig. 1) consistent with previous analyses[10,11,15] and added further details. The left end usually lies downstream of a tRNA gene that is directed toward the transposon[12,13,15]. The CRISPR array always follows in a short distance in reverse orientation, in the direction of the left end[11,13].

The CRISPR repeats in these CAST systems are most conserved in their 3′ segments[13] and 37 nt long. Next to the CRISPR array follows a tracrRNA gene (Fig. 1) on the same strand[11,25–27]. Based on the previous genome-wide mapping of transcriptional start sites (TSS)[28], we detected the tracrRNA TSS 35 nt downstream of the stop codon of the CRISPR-Cas effector gene *cas12k*, transcribed in the same direction (Supplementary Fig. 1a). The tracrRNA promoter appears to be conserved among different CAST systems (Supplementary Fig. 1b).

Next to the left end, a truncated, single repeat lies inside the transposon downstream, but usually clearly separated from the CRISPR array[13] (called R* in Fig. 1). Directly downstream of this truncated repeat R*, a truncated spacer sequence of usually 17 nt can be identified. This truncated spacer corresponds to a protospacer sequence located just outside of the transposon next to the left end (Fig. 1), usually within the tRNA gene[13]. The truncated single repeat-spacer sequences, read toward the left end, show a conserved upstream GTN-PAM, consistent with the predicted Cas12k PAM[11]. The

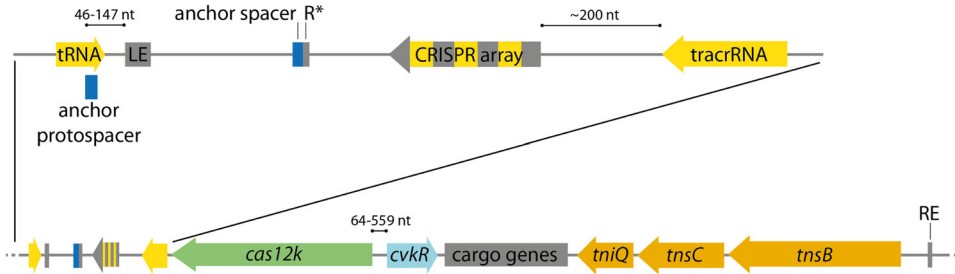

**Fig. 1 | Principal gene arrangement within cyanobacterial CAST systems.** The CAST transposon is displayed from its left end (LE) to its right end (RE). The genes are colored according to function (green: *cas12k*, orange: transposase genes, light blue: CAST regulator gene (*cvkR* in *Anabaena* 7120), yellow: regions from which non-coding RNA is transcribed, dark gray: cargo genes). On top, the region from the tRNA gene to the tracrRNA is enlarged. The CRISPR array is depicted with its repeats (gray) and spacers (yellow) separated from the 17 nt anchor spacer (blue) downstream of the array next to a truncated repeat sequence (R*; -12 nt). The scheme is not drawn to scale, but distances of particular interest or mentioned in the text are indicated.

distance from this PAM to the left end varies from 46 to 82 nt, with one exception of 147 nt (Fig. 1). Because of its conserved position at this site and experimental evidence of CAST integration via protospacer recognition[13], this motif is likely necessary for the insertion of CAST. We therefore suggest the terms anchor protospacer and anchor spacer for these sequences (Fig. 1).

Looking from the other side of the transposon, the first genes next to the right end are the three genes encoding transposase subunits TnsB, TnsC, and TniQ, always in this order[11,13,29], facing away from the right end (Fig. 1). The majority of cargo genes, located between *cas12k* and *tniQ*, are less conserved and vary in sequence and function. There is, however, one exception, a gene predicted to encode a small DNA-binding protein next to the start codon of *cas12k* in reverse orientation (Fig. 1).

## Three types of regulators are associated with the CAST systems of cyanobacteria

After identifying the 118 CAST systems and their respective left and right ends (Supplementary Data 1), we developed a database of all proteins possibly encoded within the CAST transposons with a minimum length of 50 amino acids and allowing the three start codons ATG, TTG, and GTG. We searched this database by blastP[30] for homologs of the *Anabaena* 7120 MerR-like protein Alr3614 (in the following called CvkR), previously suggested as a possible regulator[15] allowing a maximum e-value of 1e-20. We identified 53 *merR*-like genes, all located upstream of *cas12k* in reverse orientation, similar to the *cvkR* arrangement in *Anabaena* 7120.

In parallel, we used the NCBI webtool to search the conserved domain database[31] for assigning domains to the proteins in our database. Looking specifically at proteins encoded upstream of *cas12k* in reverse orientation we found ten genes with a cl10310-domain (PHA01623 superfamily) of unknown function (in the following called "Arc_1") and six genes with a cl06769-domain (RHH_5, ribbon-helix-helix; called "Arc_2" below). Further analysis with HHpred[32] predicted both sets of genes to encode small Arc-like repressors of 53 to 72 amino acids, that belong to ribbon-helix-helix DNA-binding protein superfamily[33,34]. To identify further variants, we searched for homologs of these Arc repressors within CAST systems (maximum e-value of 1e-20) and identified a total of 22 Arc_1 (alignment in Supplementary Data 2) and 11 Arc_2 genes (alignment in Supplementary Data 3), which were all located upstream of *cas12k* in reverse orientation. We found one additional Arc_1 gene that with a distance of 1.3 kb is located further away from *cas12k*, and yet another one lacking association to *cas12k*.

We further investigated this significance by searching for additional homologs to these regulators outside of the known CAST systems in NCBI and identified 210 unique genes (157 CvkR-like, 32 Arc_1-like and 21 Arc_2-like; maximum e-value of 1e-20). We searched for *cas12k* in close vicinity to these genes and found in 169 cases a *cas12k* gene or a degenerated version of it (Supplementary Table 1). We also searched for other CAST components around the regulators yielding 130 left end elements, 90 CRISPR arrays, 139 tRNA genes as well as 119 tracrRNA loci identified by using the CRISPRtracrRNA tool[27]. Just in 18 cases we could not find any CAST components close to the regulator. This high association with the CAST system supports the importance of these regulators for the transposon and *cas12k*.

Thus, three main types of possible regulators were identified, a MerR-like protein, here called CvkR, and two distinct Arc-repressors, here called Arc_1 and Arc_2.

To investigate the association of these different types of regulators with the CAST systems in different taxa, we performed a phylogenetic analysis of Cas12k proteins and mapped the respective regulators on the resulting phylogenetic tree. To avoid mistakes, we excluded likely truncated and pseudogenes, yielding 106 genes (Fig. 2, Supplementary Fig. 2, alignment in Supplementary Data 4). This analysis suggested that the three main regulator types became associated with three distinct groups of Cas12k proteins, well separated from each other. This result also suggested single separate events in which ancestral CAST systems incorporated the respective regulator. In contrast, the relationships between a particular Cas12k and a particular taxon is not congruent with the different cyanobacterial orders, likely due to horizontal transmission of CAST systems.

The CvkR family proteins range 139–185 amino acids in length, and two distinct subgroups can be recognized based on the phylogenetic analyses (Supplementary Fig. 3, alignment in Supplementary Data 5). Four CvkR proteins differed further in representing fusion proteins with an *hsdR*-restriction domain from a DNA-restriction-methylation (RM) system (Supplementary Fig. 3). RM systems are frequent among the CAST cargo genes, but there is no evidence that RM and CAST systems work together at a mechanistic level.

## Expression of CvkR from leaderless mRNA

The CvkR encoded by *alr3614* was annotated as a 168 amino acid-long MerR protein. However, sequence comparison of Alr3614 against other MerR-type CvkRs indicated that the NCBI annotation for this protein (here called CvkR-L) was too long (supported by 49 out of 53 homologs), and the actual protein would consist of only 150 residues (here called CvkR-S; Fig. 3a). Moreover, the start codon for the translation of the 150 residues CvkR-S coincides with the previously mapped TSS of its mRNA[28], strongly suggesting translation of CvkR-S from a leaderless mRNA.

In a similar fashion, we noticed that the TSS of *cas12k* coincides with the first nucleotide of a putative start codon for a reading frame yielding a 639 amino acids protein (correct in GenBank entry MBD2276705) instead of the 648 residues protein (annotated in GenBank entry BAB75312.1). Consequently, both CvkR and Cas12k are likely translated from leaderless mRNA (Fig. 3b). The intergenic spacer between the two corrected start codons and their respective TSSs is 82 nt long in *Anabaena* 7120. A generally shorter distance between these two cognate genes than with other associated regulators was also observed for other systems (Fig. 3c).

To verify the leaderless expression of CvkR-S, we constructed a deletion mutant (Δ*cvkR*) using the CRISPR-Cas12a (Cpf1) genome editing tool (Supplementary Fig. 4a, b) and three versions of complementation mutants (Δ*cvkR*Com-1, -2 and -3). For complementation, shuttle vectors were used, which carried *cvkR* genes driven by the copper-inducible *petE* promoter for the long form CvkR-L (Δ*cvkR*Com-1) or the short form CvkR-S, either containing a leader sequence (Δ*cvkR*Com-3) or not (Δ*cvkR*Com-2). For detection, a 3xFLAG epitope-encoding tag was fused to all three *cvkR* reading frames. The complementing plasmids were introduced into the Δ*cvkR* deletion background and verified (Supplementary Fig. 4c, d). The *cvkR* gene was transcribed in all three constructs (Fig. 3d). Translation of the encoded proteins was detected by Western blot, with clear size differences between CvkR-L and CvkR-S (Fig. 3e). Intriguingly, in Δ*cvkR*Com-1, not only the CvkR-L form but also a small amount of CvkR-S was detected, suggesting a propensity for initiation of translation at codon 18 even if the mRNA was 5' elongated. As there was no 5'UTR in Δ*cvkR*Com-1 following P$_{petE}$, it seems that the start codon of CvkR-S could also serve as a start site of translation if combined with transcription from a strong promoter. The incongruence between *cvkR* mRNA and protein levels within Δ*cvkR*Com-3 implied that the 5'UTR of P$_{petE}$ might impact the transcription rate, mRNA stability and/or translation efficiency of the *cvkR-S* mRNA. Taken together, we provide solid evidence that CvkR could be expressed from leaderless mRNA (Δ*cvkR*Com-2) or the start codon corresponding to codon 19 of the original annotation (Δ*cvkR*Com-3). Because the correct inducibility from the P$_{petE}$ promoter was only observed in Δ*cvkR*Com-2, we employed this strain for all subsequent complementation experiments, naming it henceforth Δ*cvkR*Com.

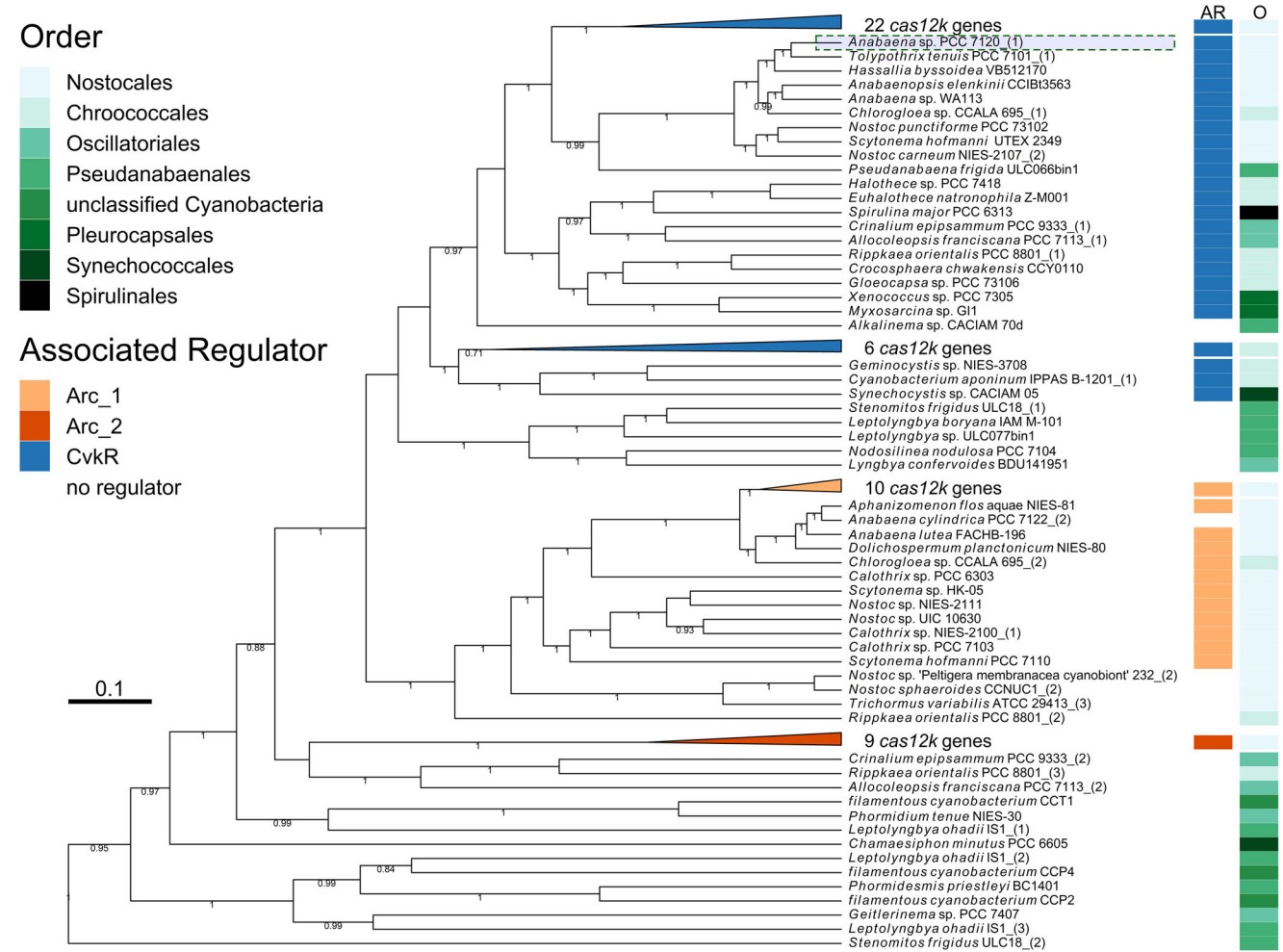

**Fig. 2 | Phylogenetic analysis of Cas12k homologs.** The identified 106 complete Cas12k proteins were aligned using M·coffee[74,75] and analyzed by BEAST[76]. Twelve instances of degenerated Cas12k proteins were intentionally left out to avoid misinterpretation. The resulting tree is depicted with branches labeled with their respective posterior probability if ≥0.7. For better recognition, the proteins were labeled with their respective host organism (further details in Supplementary Data 1). The associated regulator type (AR) and taxonomic order (O) of the respective host organism are color-coded. The *Anabaena* 7120 Cas12k (All3613) (NCBI: BAB75312.1) is marked with a green dashed box. Four nodes, whose branches were all representing *cas12k* genes with similar associated regulator genes and host organisms with similar taxonomic order, were collapsed and labeled with the respective number of *cas12k* genes. The expanded phylogenetic tree is depicted in Supplementary Fig. 2 and the multiple sequence alignment serving as source data for the phylogenetic analysis is available in Supplementary Data 4.

## CvkR impacts the expression of key components of CAST system in vivo

The AnCAST system contains all known associated genes and extends over ~21 kb (Fig. 4a). In addition to its core CAST components, it encodes multiple other transposases, possible regulators, a toxin/antitoxin pair, and an RM system. The *cas12k* gene (*all3613*) lies upstream, reverse to *cvkR*. To analyze CvkR function in vivo, the absence or overexpression of CvkR in Δ*cvkR* or Δ*cvkR*Com were first verified via Northern hybridization (Supplementary Fig. 5a) and Western blot analysis (Supplementary Fig. 5b), respectively, after induction of *cvkR* transcription in Δ*cvkR*Com from the copper-inducible *petE* promoter. Then, northern hybridization of total RNA from triplicate clones of WT, Δ*cvkR*, and Δ*cvkR*Com with single-stranded RNA probes was performed. Specific signals for *cvkR* mRNA were detected in Δ*cvkR*Com (Fig. 4b). Northern hybridizations against tracrRNA and the AnCAST CRISPR array yielded signals <500 nt, which were strongly increased in Δ*cvkR* compared to the wild-type control and decreased in intensity in the complementation strain Δ*cvkR*Com (Fig. 4c, d). We also detected an increased signal intensity for the *cas12k* mRNA in Δ*cvkR* (Fig. 4e), significantly higher than that in WT and Δ*cvkR*Com (Fig. 4f). These data provide direct evidence that CvkR directly or indirectly regulates the abundance of tracrRNA promoter-derived transcript(s) and *cas12k* mRNA.

Moreover, the presence of transcripts >400 nt indicated that tracrRNA and the CRISPR array are transcribed into a longer precursor that is subsequently processed into the major accumulating fragments of ~150 and ~210 nt, respectively. These fragment lengths are consistent with previous results of differential RNA-seq[28] and extensive transcriptome data[24] that suggested that the AnCAST CRISPR array would not have its own specific promoter. Instead, the joint tracrRNA-CRISPR array precursor is transcribed from a TSS that is located 35 nt downstream of the *cas12k* coding sequence, at position 4,362,990 on the reverse strand, 413 nt upstream of the first repeat of the CRISPR array[15] (Supplementary Fig. 1a). Nevertheless, it should be noted that the CRISPR array showed much more abundant signals in the Δ*cvkR* deletion mutant compared to WT and Δ*cvkR*Com, while tracrRNA was also well detectable in WT and Δ*cvkR*Com (Fig. 4c, d). Thus, there are additional factors involved, likely acting on the processing and stabilization of array-derived transcripts.

To investigate global transcriptomic changes upon *cvkR* deletion, microarray analysis was performed using the above verified Δ*cvkR* and Δ*cvkR*Com deletion and complementation strains (Supplementary Fig. 4). The used microarrays cover all protein-coding genes as well as noncoding RNAs and were especially designed to allow the direct hybridization of labeled total RNA without prior conversion into

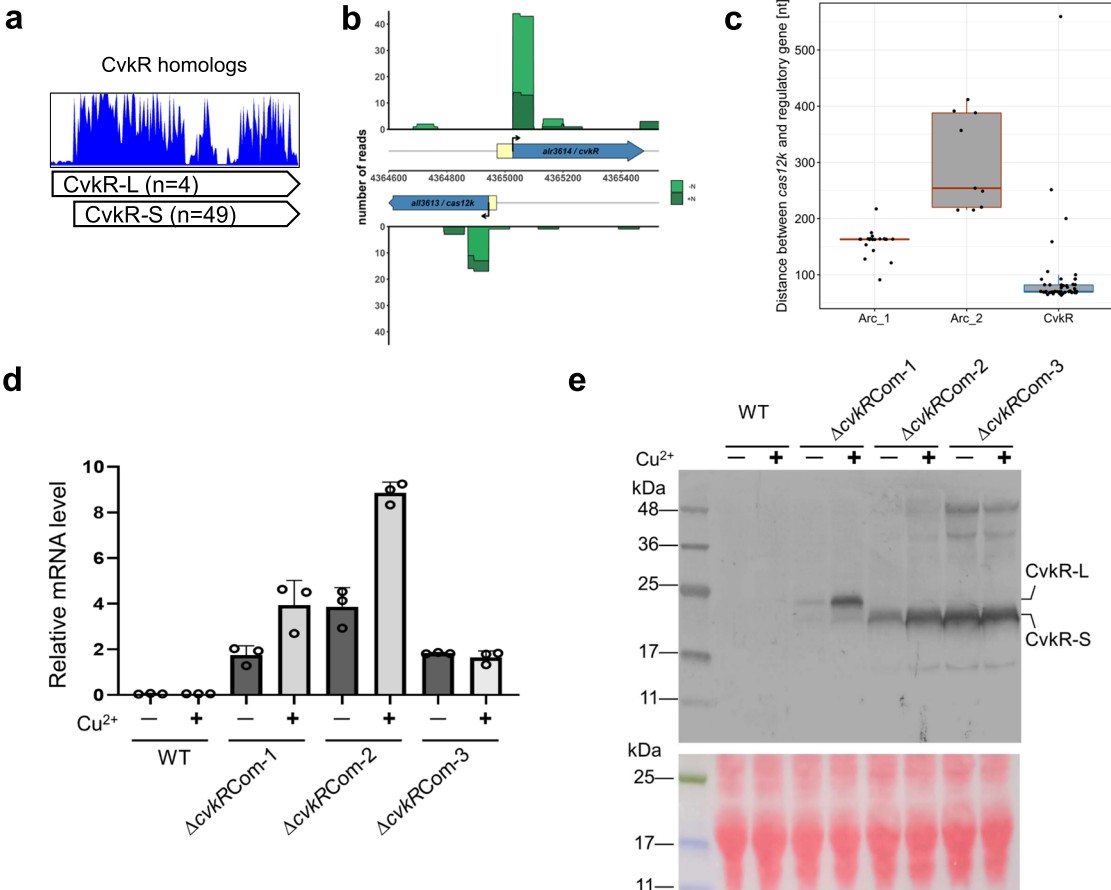

**Fig. 3 | Leaderless expression of *cvkR* genes. a** Deduced amino acid sequence alignment of 53 MerR-family CvkR homologs. The sequences of the MerR-family CvkR homologs for the multiple sequence alignment were recovered from public databases and are available in Supplementary Data 1. The sequences were aligned using MAFFT[87] and visualized with UGENE[88]. **b** Differential RNA sequencing data[28] indicate that the TSSs of *cvkR* and *cas12k* were located internal to the NCBI-annotated coding sequences under nitrogen repletion (dark green) and 8 h of nitrogen depletion (light green). Yellow boxes indicate the corresponding sections not used in translation. Data shown as primary read coverage mapped to the corresponding genomic region. Each TSS is represented by an arrow. **c** Distances between the *cas12k* effector protein and regulator genes are plotted according to the respective type of regulator, CvkR (*n* = 51 examined independent CAST systems), Arc_1 (*n* = 21) and Arc_2 (*n* = 11). The box plots are colored according to the respective DNA-binding domain (HTH: blue; RHH: red) and structured as center: median; bounds of box: upper and lower quartiles; whiskers: 1.5× interquartile range; dots: individual samples. **d** qRT–PCR analyses verify that *cvkR* is transcribed in Δ*cvkR*Com strains. The amounts of *cvkR* transcripts were normalized to those of *rnpB* as an internal standard. Data are presented as mean values ± SD (*n* = 3 biologically independent experiments). **e** Western blot analyses of different Δ*cvkR* complementation strains show expression of differently sized CvkR proteins. Upper panel, Western blot against the C-terminal 3xFLAG tag; Lower panel, ponceau S staining shows that equal amounts of protein were loaded (100 μg). The calculated molecular masses for CvkR-S and CvkR-L were 20.03 kDa and 22.41 kDa, respectively. Two independent experiments were performed, which showed consistent results. The Δ*cvkR*Com-1, 2, and 3 strains are detailed in Supplementary Fig. 4c. Source data are provided as a Source Data file.

cDNA[35]. Therefore, they enable the direct detection of tracrRNA and CRISPR array transcript levels. Microarray analysis revealed a small number of dysregulated genes. Thirteen features were significantly upregulated (seven changes in protein-coding genes and six other transcripts) and 8 were significantly downregulated (six protein-coding genes and two other transcripts) in Δ*cvkR* compared to Δ*cvkR*Com (Fig. 4g). In addition to the AnCAST genes *cas12k* and *tnsB* (*all3630*), the tracrRNA and CRISPR array were upregulated (Fig. 4h) upon deletion of *cvkR*. Hence, the role of CvkR as a regulator of the AnCAST system was confirmed and further extended.

Differentially expressed genes outside the AnCAST system included the L-array, a cryptic tRNA gene cluster relevant for survival under translational stress[36], which was upregulated here in Δ*cvkR*Com (Fig. 4g), inverse to the regulation observed for *cas12k* and *tnsB*. Other differentially regulated genes were the L-array adjacent gene *all8564* encoding an HNH-type homing endonuclease, *rtcB* encoding an RNA ligase associated with RNA repair[36] as well as *alr0739* and *alr0740* encoding a YdeI and a slipin family homolog. The effects on these genes could have been caused by the presence of erythromycin to stabilize the plasmid introduced in Δ*cvkR*Com. Therefore, the up- and downregulated genes outside of AnCAST cannot be safely associated to the CvkR regulon.

The dataset of all transcripts with meaningful fold changes is provided in Supplementary Table 2, the raw data are available from the GEO database under the accession number GSE183629 (Deletion and complementation of *alr3614* in *Anabaena* 7120).

## EMSA and TXTL assays verify CvkR repressor function in vitro

To verify the regulation of CvkR on the identified AnCAST key components, EMSAs and cell-free transcription-translation assays (TXTL)[37] were performed (Fig. 5). The purified CvkR protein appeared as a dimer, with a molecular weight of ~36.5 kDa from the size exclusion chromatography (SEC) analysis, which is approximately twofold of its theoretical molecular mass (~17.3 kDa) (Fig. 5a). This dimeric form in solution is consistent with canonical MerR-type regulators[21,38–41]. We then generated the ~80 nt long promoter fragments upstream or near the previously determined TSS[28] of *cas12k*, *tnsB*, tracrRNA, and, as negative control *psbAI* (Fig. 5b) with 5′-biotin labels and performed

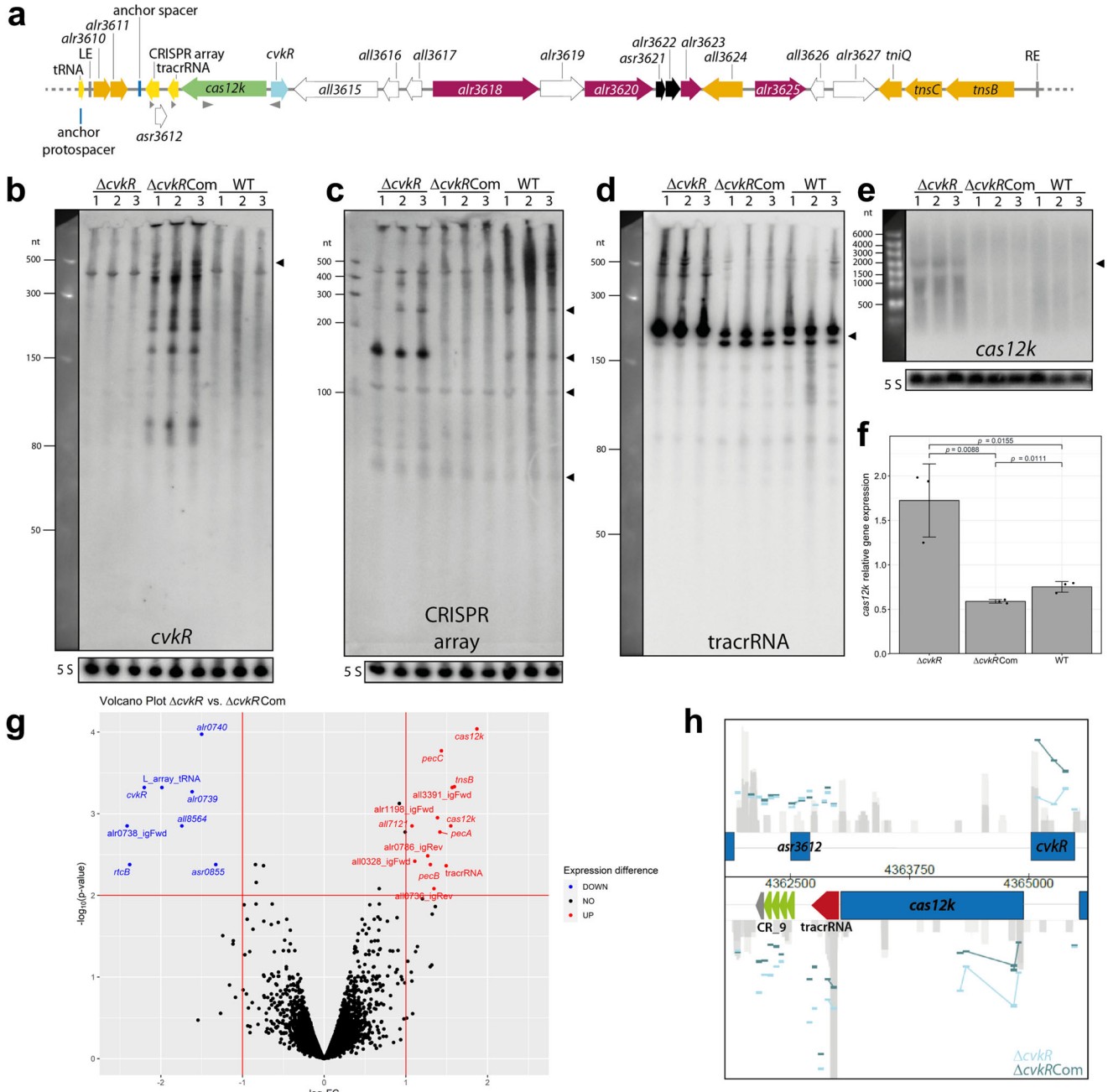

**Fig. 4 | The *Anabaena* 7120 CRISPR-associated transposase system (AnCAST) and the effects of *cvkR* deletion and overexpression. a** Genes within the 24,963 nt AnCAST include *cvkR* encoding a transcriptional regulator, *cas12k* encoding the effector, several cargo and Tn7 genes. Gray triangles indicate hybridization probes used in **b**–**e**. RM genes are colored magenta, a toxin-antitoxin module black, other gene functions as in Fig. 1. **b** to **e** Northern hybridizations for the *cvkR* mRNA, the CRISPR array, tracrRNA and *cas12k* as indicated. Triangles indicate the expected lengths for *cvkR*, for the CRISPR array (entire length ~250 nt, two repeat-spacer units ~140 nt, partially processed two repeat-spacer unit ~100 nt, single repeat-spacer 70 nt), the tracrRNA and *cas12k*, a gene of 1.92 kb. Gels and membranes were checked for equal loading by ethidium bromide staining and hybridization to the 120 nt 5S rRNA (in **b** and **d** the same membrane was hybridized). The Low Range ssRNA Ladder (NEB) was used as size marker in **b** and **d**, the RiboRuler Low and High Range RNA Ladders (Thermo Fisher Scientific) in **c** and **e**. Twenty micrograms of total RNA of WT or the respective mutants were separated on 10% PAA-urea (**b**–**d**) or 1.5% formaldehyde-agarose gels (**e**). **f** Intensities of *cas12k* mRNA signals normalized to 5S rRNA. Data are presented as mean values ± SD (*n* = 3 biologically independent samples). The two-sided, equal variance t-test showed distinct expression levels for each strain. **g** Volcano plot of Δ*cvkR* and Δ*cvkR*Com transcriptome analyses. The horizontal line marks the *p* value cutoff of 0.01, the vertical lines the fold change cutoff of |log₂| ≥1. Transcripts above these thresholds showed significant downregulation (blue) or upregulation (red) in Δ*cvkR* (details in Supplementary Table 3). **h** The most differentially expressed region in the microarray analysis belongs to the AnCAST system. Individual probes are indicated by horizontal bars, colored in light blue for Δ*cvkR* and in dark blue for Δ*cvkR*Com. Analyses shown in **b** to **e** were performed twice with consistent results. Source data are provided as a Source Data file.

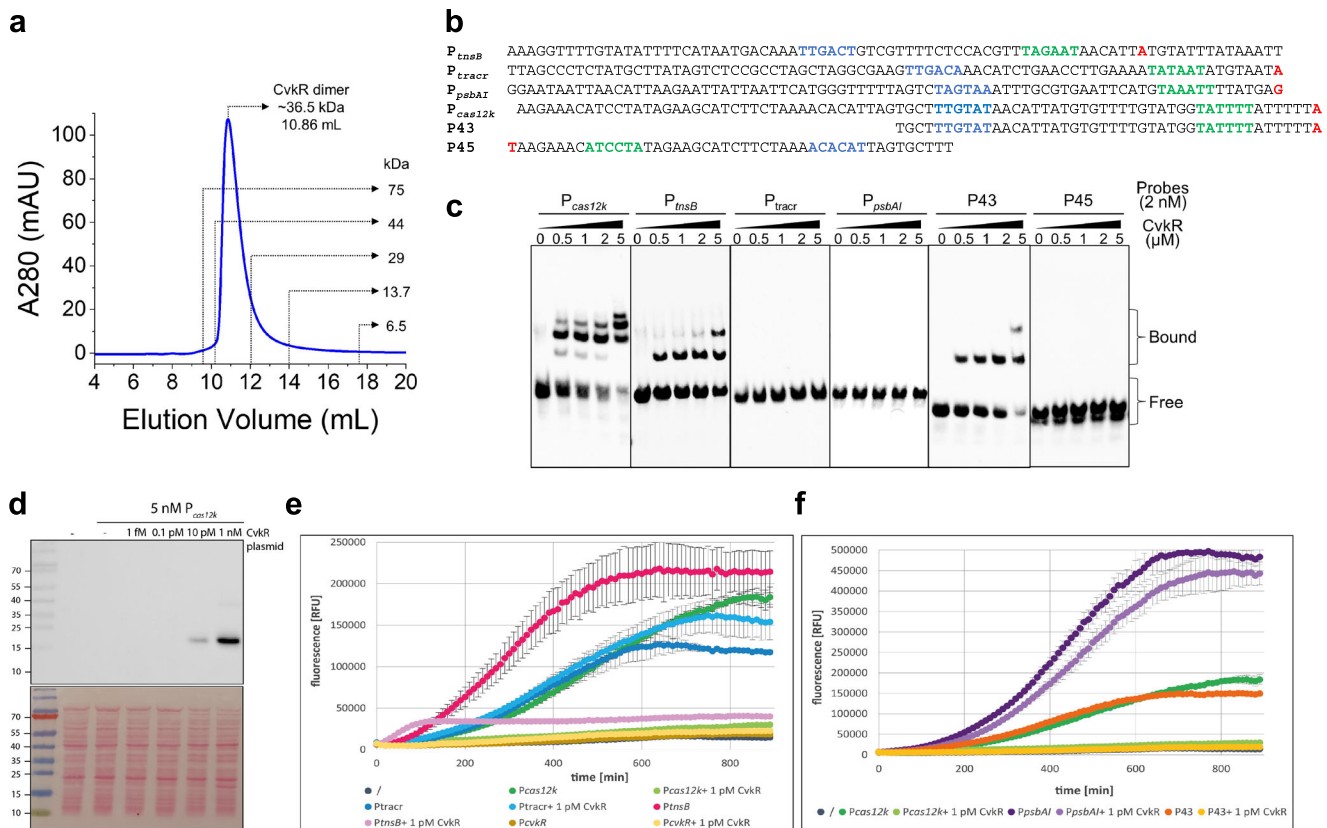

**Fig. 5 | Assays to test *cas12k* promoter elements. a** Molecular weight estimates of CvkR on size exclusion chromatography (SEC) with Superdex 75 10/300 GL column (GE Healthcare). Calibration standards are indicated. The elution volume of CvkR was ~10.86 mL, corresponding to a mass of ~36.5 kDa (twice its calculated molecular weight). mAU, milliabsorbance units. **b** Sequences of the tested promoter fragments. The probe P45 indicates the reverse complementary sequence of *cvkR* promoter. Putative −35 and −10 of *cas12k, tnsB*, tracrRNA, *psbAI* and *cvkR* promoter elements are shown in blue and green, respectively. The TSSs were included and marked in red. **c** EMSA assays showing the binding ability of CvkR to P*cas12k*, P*tnsB*, P*tracr*, P*psbAI* and truncated fragments (P43 and P45) of P*cas12k*. The data are presentative from two or three independent experiments. The free probe and complexes of protein-DNA complexes were marked as "Free" and "Bound", respectively. **d** CvkR was expressed from vector pET28a and was detected by Western blot analysis via the N-terminal 6xHis tag (upper panel), and the stained membrane is shown in the lower panel (one out of three independent experiments is shown). The corresponding size for 6xHis-CvkR is 19.9 kDa. The prestained PageRuler (Thermo Fisher Scientific) was used as a size marker. **e** TXTL assay[37] to compare the promoter activities of *cas12k*, *cvkR*, *all3630* (*tnsB*) and tracrRNA (P*cas12k*, P*cvkR*, P*tnsB*, and P*tracr*) to drive deGFP expression in the absence or presence of CvkR. **f** The full-length version of P*cas12k* and the P43 fragment encompassing 43 nt upstream of the *cas12k* TSS were tested in the TXTL system for their capacity to drive deGFP expression and mediate repression upon parallel expression of CvkR. P*psbAI* was used as negative control for a promoter not controlled by CvkR. In both **e** and **f**, 1 pM CvkR plasmid was expressed together with the corresponding p70a plasmids (5 nM) with the respective promoter variants upstream of deGFP. Error bars show standard deviations calculated from two technical replicates in one representative example of three independent experiments. Source data are provided as a Source Data file.

EMSA assays with recombinant CvkR at different concentrations. The results confirmed that CvkR directly binds to P*cas12k* as well as P*tnsB* but not P*psbAI* (Fig. 5c). Surprisingly, there was no binding between CvkR and P*tracr* (Fig. 5c), despite the drastically changed abundances of tracrRNA-CRISPR array transcripts in the Δ*cvkR* mutant in vivo (Fig. 4c, d). To independently verify these results, we cloned the respective promoters upstream of a deGFP reporter gene to test whether they can drive transcription and thereby deGFP production in the TXTL assay[37]. In parallel, CvkR was expressed from a second plasmid (Fig. 5d). The P*cas12k*, P*tnsB*, P*psbAI* and P*tracr* promoters were functional and able to drive the expression of the deGFP gene. In the presence of CvkR, deGFP production was repressed in reactions containing plasmids with deGFP driven by P*cas12k* and P*tnsB* but not P*tracr* and the negative control P*psbAI* (Fig. 5e, f). These results are in good agreement with the gel shift assays. We conclude that the increased *cas12k* and *tnsB* mRNA levels in Δ*cvkR* were a direct consequence of the *cvkR* deletion, whereas the higher accumulation of P*tracr*-derived transcripts must have resulted from indirect CvkR regulation. The promoter of *cvkR* (P*cvkR*) was not able to drive deGFP transcription in the TXTL system (Fig. 5e). Therefore, it could not be tested whether CvkR can regulate its own transcription. If the overlapping *cas12k* and *cvkR* promoters were

separated into two fragments, we did not observe CvkR binding to the P45 fragment containing the core region of P*cvkR* including the predicted −10 and −35 elements in the EMSA assay. In contrast, strong binding of CvkR was observed to the P43 fragment containing the core region of P*cas12k* (Fig. 5b, c). These results point away from a possible autoregulation of CvkR.

## Identification of CvkR binding site

To further define the minimal necessary sequence and motifs required for CvkR binding, P*cas12k* and P*tnsB* were analyzed in DNase I footprinting assays (Fig. 6a). CvkR specifically protected a 43-nt fragment of P*cas12k* (P*cas12k*center) and a 41-nt fragment of P*tnsB* (P*tnsB*center). Intriguingly, a nearly perfectly conserved 16-nt sequence (5′-ATAACATTATGTRTTT-3′) was shared between the protected regions of P*cas12k*center and P*tnsB*center (Fig. 6a). Thus, this sequence is at the core of a potential CvkR binding motif. Bioinformatic screening of CAST systems closely related to AnCAST (Supplementary Fig. 3) identified a 15 nt inverted repeat within this sequence (5′-<u>ATAAC</u>ATT<u>ATGTGTT</u>-3′). This potential CvkR motif is widely conserved in the promoter regions of *cas12k* and *tnsB* homologs in cyanobacteria (Fig. 6b and Supplementary Fig. 6).

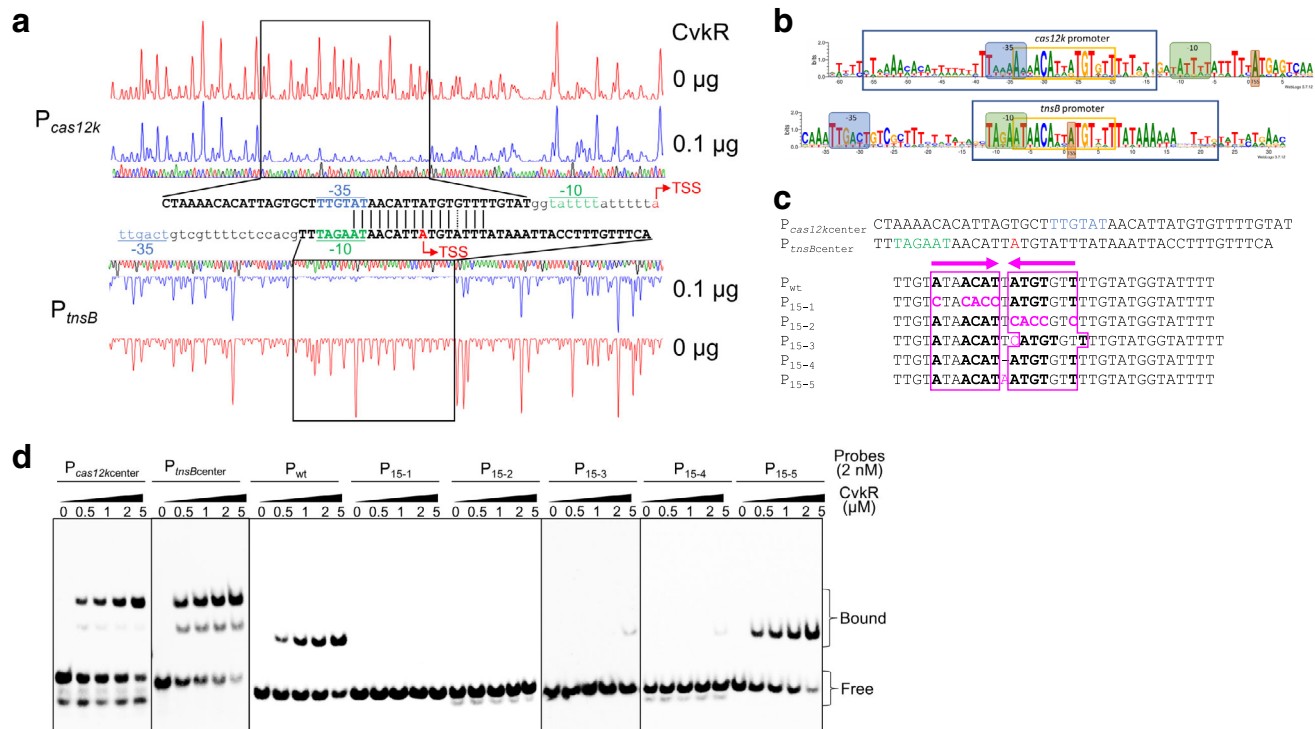

**Fig. 6 | Analysis of CvkR binding motifs. a** DNase I footprinting of the FAM-labeled $P_{cas12k}$ and $P_{tnsB}$ promoter sequences in the absence (red graphs) and presence of 0.1 μg CvkR (blue graphs). The protected region is boxed. Dideoxynucleotide sequencing traces are displayed (green, A; blue, C; black, G; red, T) and the DNA sequences protected by CvkR are indicated below (for $P_{cas12k}$) or above (for $P_{tnsB}$) the corresponding regions in capitalized bold letters. The sequence overlap is marked by vertical black lines. A dashed line indicates a single different nucleotide. The area adjacent to the CvkR-protected regions is displayed in lowercase letters. The TSS (red), −10 (green) and −35 elements (blue) are highlighted. **b** Sequence logo of the CvkR motif based on sequence alignments of $cas12k$ and $tnsB$ promoters of 15 CAST systems with $cvkR$ genes (for sequence details see Supplementary Fig. 6). In the sequence logos, the −35 and −10 promoter regions are shaded in blue and green, and the TSSs in brown. The protected areas in the DNase I footprinting assays are boxed in blue and the palindromic CvkR binding motif in orange. **c** Wild-type and mutated promoter fragments used for EMSA assays. The probes $P_{cas12kcenter}$ and $P_{tnsBcenter}$ indicate the CvkR-protected region of $cas12k$ and $tnsB$ promoters in the DNase I footprinting assay, respectively. TSS, −10, and −35 elements in $P_{cas12kcenter}$ and $P_{tnsBcenter}$ are indicated as in **a**. CvkR-binding sites are pink-boxed. The inverted repeat is displayed in bold letters and highlighted by the pink arrows, while pink letters indicate the mutated sites. **d** Interaction between CvkR and $P_{cas12k}$ variants analyzed by EMSA. The data are presentative from two or three independent experiments. The free probe and complexes of protein-DNA complexes are marked as "Free" and "Bound", respectively. Source data are provided as a Source Data file.

To validate the proposed CvkR-specific binding motif, we carried out EMSA assays with probes in which different nucleotides were substituted within the palindrome (Fig. 6c). These assays verified the interaction between CvkR and probes $P_{cas12kcenter}$ and $P_{tnsB}$center (Fig. 6d) and that CvkR binds to $P_{wt}$ in a similar concentration-dependent manner. When the reverse complementary sequence in this motif was mutated ($P_{15-1}$ and $P_{15-2}$) or the spacer length was changed ($P_{15-3}$ and $P_{15-4}$), CvkR binding was nearly abolished. However, merely changing the spacer nucleotide from T to A without changing the length had no effect ($P_{15-5}$) (Fig. 6c, d). The results provided strong evidence that CvkR binds to a 15-nt motif 5′-AnnACATnATGTnnT-3′, and that this type of CvkR-DNA binding is widely distributed among the CAST systems of cyanobacteria.

### The crystal structure reveals that CvkR represents a unique type MerR-type regulator

To investigate CvkR functionality at the structural level, a high-quality crystal structure was determined at 1.6 Å resolution. CvkR consists of a classical winged helix-turn-helix (wHTH) DNA-binding domain at the N-terminus (residues 1-80), a dimerization domain (residues 81-132), and a potential effector-binding helix in the C-terminal region (helix α7, residues 133-150), in which an ATP, used as reagent for optimal crystal formation, was bound as a ligand (Fig. 7a). The N-terminally located wHTH domain with an α1-α2-W1-α3-W2-α4 topology is a typical feature of MerR-family

regulators[21,38–42], and therefore indicates that CvkR belongs to this family from a structural perspective.

The members of this family typically adopt a physiologically active homodimeric form[21,22,39,40,43–46]. By dynamic light scattering (DLS), a ~5.1 nm (~51 Å) average diameter was measured for CvkR (Fig. 7b), consistent with its assembly into a homodimer of ~54 Å in the solved crystal structure (Fig. 7c and Supplementary Fig. 7) and the result of our SEC analysis (Fig. 5a). Three antiparallel β-strands (β2-β1-β3) and one short α-helix (α6), together with the equivalent β-strands and helix of the other subunit, comprise the bulk of the dimerization interface (Fig. 7c). The contacts are predominantly hydrophobic (Fig. 7d) and bury 997.6 Å² surface area (PDBePISA)[47] (Supplementary Fig. 7), which is much smaller than that observed in canonical MerR-type proteins, for instance, 1896.8 Å² in CueR. In comparison, the mode of CvkR dimerization significantly differs from MerR-type regulators such as CueR[39] or MtaN[48], which assemble into homodimers via two-helix antiparallel coiled-coil (α5/α5′) (Fig. 7e), and is also distinct from the unusual MerR-type TnrA/GlnR family regulators[39,40], which form homodimers relying on an N-terminal extra helix (e-α1/e-α1′) (Fig. 7f). Therefore, the pattern of CvkR dimerization is unique among the MerR family members.

Another striking feature in the CvkR structure is that the C-terminal four-turn α helix (α7) packs against its own wHTH DNA-binding domain (α1 and α3) and dimerization domain (η1 to α5), but not against these domains of its dimeric partner as in other MerR-type

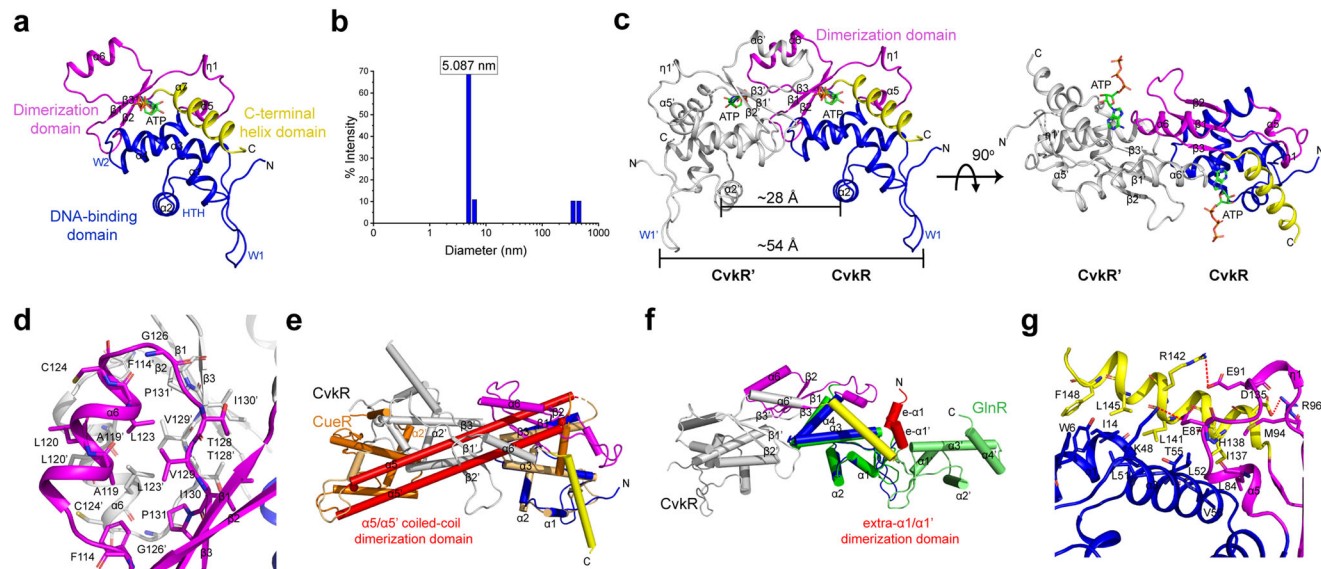

**Fig. 7 | Structural analyses of CvkR. a** Overall structure of the CvkR-ATP complex. The DNA-binding, dimerization, and putative effector-binding C-terminal helix domains are shown in blue, magenta, and yellow, respectively. The typical helix-turn-helix (HTH) and two "wing" loops W1 and W2 in the DNA-binding domain are indicated. The ATP ligand is represented in green stick and colored by the atom type. **b** Dynamic light scattering analysis of the average diameter of native CvkR protein measured at 25 °C. The data are representative from three independent experiments. **c** Cartoon representation of the CvkR homodimer. The other subunit is shown in white. The bulk of the dimerization interface is formed by three anti-parallel β-strands (β2-β1-β3) and one short α-helix (α6) from the two subunits in the dimer. The distances between two DNA-interacting helices α2/α2' and between two W1 loops W1/W1' are measured and labeled, respectively. **d** The dimerization interface mainly consists of hydrophobic interactions. The key residues involved in homodimer formation are shown in sticks and labeled in black. **e, f** Comparison of dimerization patterns in MerR-type proteins. Superimpositions are performed with reference to the wHTH domain from one subunit in homodimers. Helices are shown as cylinders for clarity. Superimposed proteins: CvkR (this work; coloring as in **c**), CueR (PDB code 1Q07, two subunits coloring as orange and light orange, respectively, effector Au showing in red sphere), GlnR (PDB code 4R4E, two subunits coloring as green and lime, respectively). The dimerization domains for CueR and for GlnR are colored red. **g** Close-up view of the interactions for C-terminal helix α7 packing. The key residues are shown in sticks and three salt bridges are marked by red dash lines.

regulators such as CueR or SoxR[39,40] (Fig. 7a, e). Three specific salt bridges and hydrophobic interactions contribute to the precise orientation of helix α7 (Fig. 7e, g).

MerR-like proteins with a recognizable C-terminal sensor domain, such as metal-responsive CueR, MerR and redox-responsive SoxR[39,40,49], possess cysteine residues for effector binding at conserved positions. However, the three cysteines present in CvkR are not near the potential effector-binding helix α7 (Fig. 7g and Supplementary Fig. 8). This suggests that CvkR might be one of those MerR-like proteins that lack a recognizable C-terminal sensor domain, like HiNmlR from *Haemophilus influenzae*[42]. In our solved structure, the crystal optimizing reagent ATP was bound near the putative effector binding site through π−π stacking, cation−π, and several hydrogen bonds (Supplementary Fig. 9a). While the nonspecific hydrogen bonds at the N1 and N6 positions of the adenine moiety, together with the unbound triphosphate group in the putative effector binding site, suggest that ATP is not the natural effector of CvkR, other nucleoside derivatives might be possible candidates (Supplementary Fig. 9b). Indeed, signaling molecules of the cyclic oligonucleotide family have been observed in certain types of CRISPR-Cas and other defense systems[50,51]. However, subsequent EMSA experiments showed no effects on CvkR DNA binding when such ligands were added, indicating that ATP and its analogs are indeed not the natural effectors of CvkR (Supplementary Fig. 10).

**Proposed molecular basis of CvkR DNA binding**

To further elucidate the molecular basis of CvkR DNA binding, we aimed at solving the CvkR-promoter complex structure but reached no better diffraction resolution than 6 Å, insufficient for structure determination. Considering the relatively conserved DNA-binding characteristics of the wHTH domain among MerR-family members[21,40,42], we resorted to analyzing the binding of CvkR to DNA by structural alignment. CvkR shares the lack of a recognizable C-terminal sensor region and the fact that overlapping promoters can be targeted with HiNmlR[42]. Therefore, the HiNmlR-DNA complex (PDB code 5D8C) was chosen for comparison. Due to the shorter distance between the two DNA-interacting helices α2 and α2' of CvkR, only one wHTH domain was matching well (Fig. 8a). Consistent with the DNA binding of wHTH domains, on the well-matched side, helix α2, which is rich in positively charged residues, and winged loop W1 separately insert into the major and minor grooves of DNA, while winged loop W2 is close to the DNA backbone (Fig. 8a, left panels). Based on the comparison to HiNmlR, residues R20, Q23, and Y24 in α2, and residue N66 in W2 were suggested as possibly involved in the interactions between CvkR and its target (Fig. 8a, right panel)[42]. Moreover, the positively charged residues R19, K40 and R42 in helix α2 and W1 are also possibly participating in DNA binding, especially R42, whose side chain simultaneously forms hydrogen bonds with the ATP's α-phosphate group (resembling the phosphate in the DNA backbone) in our solved structure (Fig. 8a and Supplementary Fig. 9b).

To address the relevance of these residues for target promoter binding, two mutated proteins were constructed, CvkRmut (R19A-R20A-Q23A-K40A-R42A-N66A) and R42E. As observed with wild-type CvkR (Fig. 5a), both mutated proteins appeared in dimer form in the SEC (Supplementary Fig. 11) suggesting that they were correctly folded and competent for biochemical analysis. The subsequent EMSA experiments showed for both proteins no shifted bands compared to wild-type CvkR (Fig. 8b). Thus, both CvkRmut and R42E had lost the capability to specifically bind P$_{cas12k}$, directly supporting a role of the substituted residues in DNA binding.

To verify the role of these residues in transcriptional regulation by CvkR, TXTL assays were performed. Western blots showed that the

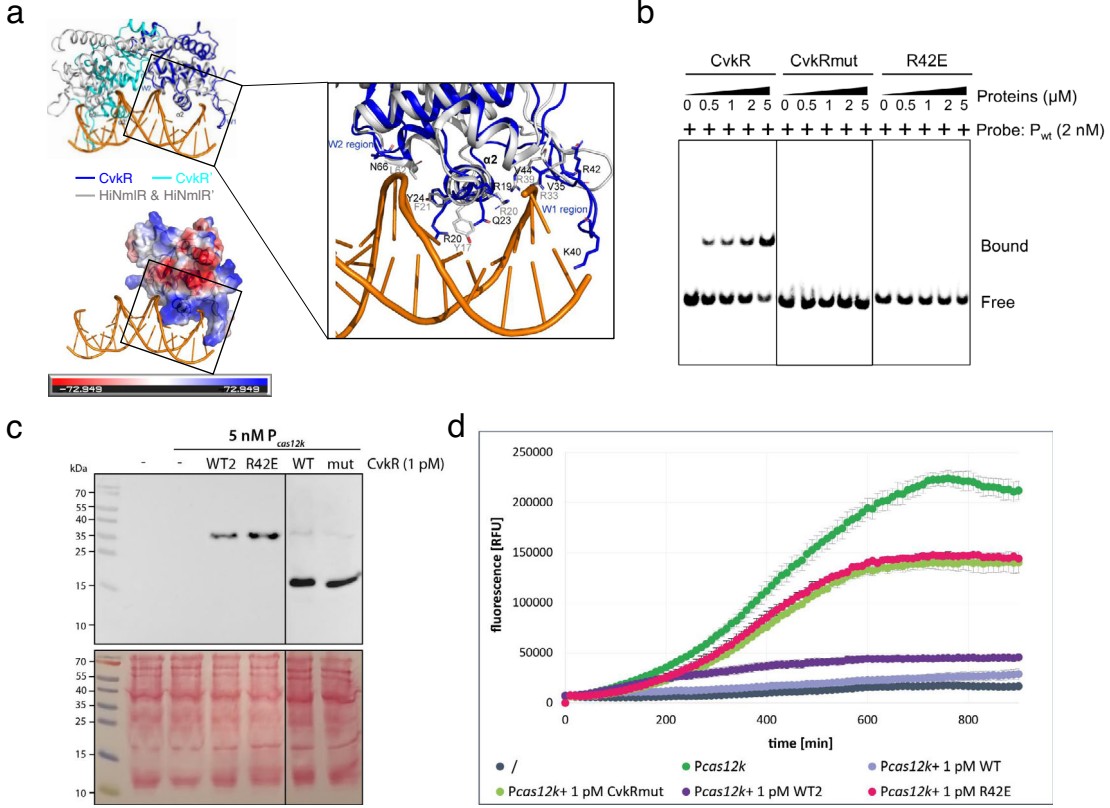

**Fig. 8 | Functional characterization of CvkR DNA-binding residues. a** Upper left panel: Superimposition of CvkR onto the HiNmlR-promoter complex. Two subunits of the CvkR homodimer are colored in blue or cyan. HiNmlR homodimer and its target DNA in the HiNmlR-promoter complex (PDB code 5D8C) are displayed in gray and orange, respectively. Three regions directly participating in promoter DNA binding in the HiNmlR-promoter complex, helix α2 and two winged loops W1 and W2, are labeled. Lower left panel: electrostatic surface representation of CvkR highlighting the potential electropositive (blue) DNA-binding surface. Right panel: The key DNA-interacting residues reported in HiNmlR are shown in sticks and labeled in gray. The possible DNA-interacting residues in CvkR corresponding to the three DNA-interacting regions are labeled in black. **b** EMSA assays showing the binding of CvkR variants to $P_{cas12k}$. The 33-nt $P_{wt}$ probe was used for the assay. The His-SUMO tags were removed from all CvkR variants used in the EMSA assays. The free probe and $P_{wt}$-protein complexes are marked as "Free" and "Bound", respectively. The data are representative from three independent experiments. **c** CvkR

and mutant proteins were expressed in the TXTL system[37] from pET28a(+) or pET28a(+)-SUMO and detected by Western blot via their N-terminal 6xHis tag. The stained membrane is shown below. The corresponding molecular masses are 29.9 kDa for 6xHis-SUMO-CvkR WT2 and R42E, while 6xHis-TEV-CvkR WT and 6xHis-TEV-CvkRmut have masses of 19.9 and 19.5 kDa. **d** TXTL assay to test the regulatory capacity of CvkRmut and R42E compared to CvkR (WT and WT2 as defined in panel C) for repressing deGFP fluorescence expressed from the *cas12k* promoter ($P_{cas12k}$). The $P_{cas12k}$-deGFP cassette was expressed from plasmid p70a (5 nM) in the absence or presence of 1 pM pET28a(+) expressing CvkRmut, R42E, CvkR WT or WT2. The TXTL reactions were performed at 29 °C overnight, and fluorescence was measured every 10 min in a Wallac 1420 Victor2 microplate reader. Error bars show standard deviations derived from two technical replicates in one representative example of three independent experiments. Source data are provided as a Source Data file.

CvkR wild type protein was stably produced both without (WT) or with a SUMO tag (WT2) in the TXTL system and that the CvkR mutant proteins with SUMO (R42E) or 6xHis (CvkRmut) tag were produced as well (Fig. 8c). When CvkRmut and R42E were coexpressed with the reporter plasmid, almost no inhibiting effect on $P_{cas12k}$ compared to the CvkR controls was observed (Fig. 8d). These observations were consistent with the results of the EMSA assays and supported our model for DNA interactions by CvkR (Fig. 8a, right panel).

## Discussion

Native type V-K CAST systems have thus far only been found in certain cyanobacteria[11,15]. Their better characterization is of fundamental interest, and these systems hold great promise for the development of useful genome editing tools[52]. The primary function of native CRISPR-Cas systems is defense against mobile genetic elements. Therefore, we reasoned that the activity of CAST systems that can transpose to different sites within a genome must be controlled. Here, we addressed the association between putative transcriptional regulators and cyanobacterial CAST systems and found three different types of such repressors. We suggest to name one of them CvkR, for Cas-type V-K

repressors, while the other two were called here Arc_1 and Arc_2 due to the lack of a better classification. The consistent occurrence of these regulators in 86 of 118 analyzed systems indicates a functional dependency. Judged by the diversity of regulators and their DNA-binding domains, it is likely that they were acquired several times independently.

Both *cvkR* and *cas12k* were found to be expressed from leaderless mRNAs. Compared to eukaryotes and archaea, the frequency of leaderless mRNAs in bacteria, especially in gram-negative bacteria, is rather low[53–55]. The dRNA-seq analysis of transcriptomes from 14 different growth conditions provided evidence for 51 leaderless mRNAs in the cyanobacterium *Synechocystis* 6803[56]. Even in the extensively studied *E. coli*, deep sequencing found less than 30 leaderless mRNAs under standard growth conditions[57]. As such, the biological function of leaderless gene expression largely remains to be established. Tn1721 *tetR* mRNA and λ*cI* mRNA are the best-studied leaderless mRNAs derived from transposons and bacteriophages in *E. coli*. These leaderless mRNAs were proposed as mediators of horizontal gene transfer which are controlled at the translational level[54] and this might apply to CvkR as well, especially as it was identified in this study as

regulator of the CAST system, which has ties to transposons and phages. However, the real relevance of leaderless mRNA translation in cyanobacterial CAST systems needs to be further studied.

We then characterized CvkR, the transcriptional regulator of the AnCAST system in detail. The amounts of *cvkR* mRNA and CvkR protein in wild-type cells were very low under the tested growth conditions. However, *cvkR* deletion led to clearly enhanced transcript levels of *cas12k*, *tnsB* and tracrRNA in the Δ*cvkR* mutant. Therefore, CvkR likely is expressed at a low, but sufficient level to perform the repressor function in *Anabaena* 7120 wild type. We elucidated that CvkR controls P*cas12k* and P*tnsB* directly and indirectly the abundance of the tracrRNA-CRISPR array. We reasoned that the increased accumulation of tracrRNA might result from protection through binding to the Cas12k effector that is present in higher amounts in Δ*cvkR*. All these results point at the biological function of CvkR as a transcriptional repressor of the AnCAST core module (i.e., tracrRNA, *cas12k* and *tnsB* mRNAs).

MerR-type regulators typically contain a conserved N-terminal wHTH domain[22,58]. They form homodimers and bind a palindromic DNA motif[49]. The key residues that are directly involved in interaction with DNA are conserved among members of the MerR family that recognize the same DNA motifs but these residues vary among members that recognize different DNA motifs[59]. By structural superimposition (Fig. 8a), we tentatively defined CvkR residues that participate in the interaction with DNA. The functionality of R42 and additional residues was confirmed in both, EMSA and TXTL assays (Fig. 8b, d). Based on further multiple sequence comparison (Supplementary Fig. 12a), we propose the positively charged and highly conserved R42 to directly interact with the DNA phosphate backbone.

The palindromic DNA motifs recognized by MerR-family proteins are usually located in promoters with an elongated distance (≥19 nt) between the -10 and -35 elements. Consequently, several MerR-family members employ a DNA distortion mechanism for regulating the expression of their target genes[22,59], such as the metal-responsive (i.e., MerR[43], ZntR[44], and CueR[39,45,46]), the redox-responsive (i.e., SoxR[40] and HiNmlR[42]) and antibiotic resistance (i.e., BmrR[21]) transcriptional regulators. In this study, we performed DNase I footprinting assays with both P*cas12k* and P*tnsB*. We obtained CvkR-protected regions with similar distances and a 16 nt long sequence shared between the two promoters. Targeting this region, we eventually confirmed the CvkR motif with a sequence of 5'-**A**nn**ACAT**n**ATGT**nn**T**-3' assisted by a series of EMSA assays and bioinformatic analysis (Fig. 6). When predicting the −10 and −35 elements of P*cas12k* with PromoterHunter[60], the 21 nt long spacer separating the -10 (TATATT) and −35 (TTGTAT) boxes was clearly supported. Because the CvkR motif mainly locates within this spacer, CvkR might modulate P*cas12k* activity using the MerR-specific DNA-distortion mechanism. However, for P*tnsB* we could only find a −10/−35 spacer of 17 nt (Fig. 6b). As we demonstrated that CvkR binding in P*tnsB* overlaps the TSS of P*tnsB*, and partially the -10 element, CvkR should regulate the expression of *tnsB* in a different way from *cas12k*. Actually, a few MerR-type regulators indeed have been reported to function as transcription repressors and regulate gene expression in significantly different ways, such as BldC[61], HnoC[62], and McdR[63]. BldC binds DNA direct repeats as cooperative multimers to regulate gene expression, while both HnoC and McdR are instances of regulators recognizing palindromic motifs overlapping TSS and -10 elements. We even noticed a 20-nt long −10/−35 spacer in the promoter of one of the HnoC targets, *hnoD*, suggesting a possible similar regulatory mechanism of interfering with association of the RNAP holoenzyme and transcription initiation rather than reshaping of promoter DNA[62]. McdR, another MerR type repressor involved in controlling cell division, DNA repair, drug resistance and other stress responses in Mycobacteria, could also repress target promoters in this manner[63]. Moreover, the here investigated cyanobacterial CvkR shows 33% sequence identity with mycobacterial McdR (Supplementary Fig. 13a) and possesses structural features very similar to the

alphafold2-predicted McdR model (Supplementary Fig. 13b), implying similarities in the regulatory mechanism.

Different from the N-terminal DNA binding domain of MerR-type regulators, the C-terminal effector binding domains are not conserved and vary widely in sequence and length. Multiple sequence alignment showed one of the otherwise widely conserved cysteines is in CvkR replaced by a serine (position 134, Supplementary Fig. 12a). Therefore, we generated and tested a S134C mutant protein, but it showed in SEC, EMSA, and TXTL experiments very similar characteristics to the CvkR wild type protein (Supplementary Fig. 12b–d). Structural analysis also showed that CvkR lacks the conserved cysteines for effector binding on the scaffold constructed by dimerization domain and C-terminal domain (Fig. 7d). Moreover, the cysteines of CvkR are distributed structurally dispersed, and are arranged far away from the putative effector binding site based on the structural superimposition (Supplementary Fig. 8). These findings imply the specific effector binding of CvkR likely differs from the conventional cysteine-dependent manner.

HiNmlR, lacking the recognizable C-terminal sensor, modulates responses against formaldehyde challenges by modification of two conserved cysteine residues in its wHTH domain (C54 in α3 and C71 in α4)[42]. Two conserved cysteine residues C17 and C70 are also found in the CvkR wHTH domain, and these interact with the helix α3 in our solved structure (Supplementary Fig. 8). Besides, we found that helix α3 simultaneously interacts with DNA-interacting helix α2 and the C-terminal helix α7, suggesting a possible structural rearrangement-sensing mechanism that could be triggered by modification of cysteine residues. In addition, it is also possible that CvkR binds an effector protein instead of a metabolite to conduct the transcriptional regulation, like GlnR that utilizes glutamine synthetase as its effector in gram-positive bacteria[64].

To summarize, we identified and characterized CvkR as the regulator of the AnCAST system. Structural analysis of CvkR revealed both unique dimerization and ligand binding domains. The effector molecule and the functionality of this hypothetical signaling input sensed by CvkR are matters of further research.

## Methods

### Cultures of cyanobacteria and construction of mutant strains

*Anabaena* 7120 and its derivatives were grown photoautotrophically in BG11 liquid medium or on agar plates under white light illumination of 30–50 μmol photons m$^{-2}$ s$^{-1}$ at 30 °C[65]. In terms of the controls in copper-inducible experiments, transparent plastic tissue culture flasks were used for cultivation, deionized water was used for medium preparation, and three resuspensions with copper-free medium were employed for the seed culture. In addition, excess CuSO$_4$ (1.0 or 1.25 μM) was added to guarantee P*petE* activity for the respective experiments. The culture was supplemented with erythromycin (10 μg mL$^{-1}$) when necessary.

For construction of *Δalr3614* (Δ*cvkR*), the CRISPR-Cas12a (Cpf1) genome editing tool together with the pSL2680 plasmid (Addgene No. 85581) were used[24]. The primer pair alr3614gRNA-1/2 was used to prepare the gRNA-cassette editing plasmids, and the primer pairs alr3614KO-1/2 and alr3614KO-3/4 were used to prepare the gRNA & repairing-cassette editing plasmids. The primer pairs alr3614-3/4 were used to check the deletion genotype. A 200 bp internal fragment (4365123-4365322) of *cvkR* was ultimately deleted.

For complementation of the Δ*cvkR* mutant and/or verification of leaderless expression of the *cvkR* gene, three cassettes (P*petE*_no 5'UTR-*cvkR*-L-3xFLAG, P*petE*_no 5'UTR-*cvkR*-S-3xFLAG and P*petE*-*cvkR*-S-3xFLAG) were cloned into a shuttle vector (pRL59EH) derived from the broad-host-range plasmid RSF1010 by seamless assembly. The primer pairs 59M-F/59M-R1, P*petE*-F/P*petE*-R4 and 3614L-F/3614-R were used to construct a complementation plasmid to generate Δ*cvkR*Com-1. The primer pairs 59M-F/59M-R1, P*petE*-F/P*petE*-R5 and 3614S-F1/3614-R were

used to construct a complementing plasmid to generate Δ*cvkR*Com-2. The primer pairs 59M-F/59M-R1, P*petE*-F/P*petE*-R6 and 3614S-F2/3614-R were used to construct a complementing plasmid to generate Δ*cvkR*Com-3. Each complementing plasmid was introduced into the Δ*cvkR* mutant by conjugal transfer[66]. Genotypes of mutants were confirmed by PCR (Supplementary Fig. 4d). The sequences of all oligonucleotides are listed in Supplementary Table 3. All PCR fragments, plasmids generated in this study, and gene mutation regions in the mutants were verified by Sanger sequencing.

## Microarray analysis

*Anabaena* 7120 strains Δ*cvkR* and Δ*cvkR*Com were grown in 50 mL BG11 without CuSO₄ to an OD₇₅₀ of 0.8, and CvkR expression was induced from P*petE* with 1.25 μM CuSO₄ for 24 h. Cells were harvested by centrifugation, and following resuspension, cells were transferred to screw cap tubes together with 250 μL glass beads (0.1–0.25 μm in diameter), 1 mL PGTX[67] was added and tubes were frozen in liquid nitrogen. Cell disruption (3 cycles of 3 × 6500 rpm for 15 s with 10 s breaks in between) with Precellys 24 Dual homogenizer (Bertin) was performed under cooling with liquid nitrogen. Samples were separated from the beads and incubated for 10 min at 65 °C in a water bath, one volume chloroform:isoamyl alcohol (24:1) was added, and samples were incubated for 10–30 min at room temperature with several vortexing cycles. After centrifugation for 3 min at 3250 *g* in a swing-out rotor, the supernatant was transferred to a fresh tube, and one volume of chloroform:isoamyl alcohol was added. This step was repeated twice. RNA was precipitated by the addition of one volume of isopropanol.

The RNA samples of two biological replicates each were hybridized to 8x44K microarrays (Agilent ID 062842) following published sample preparation and hybridization details[68]. In short, 2 μg of DNase-treated RNA was used for Cy3 labeling (ULS Fluorescent Labeling Kit for Agilent Arrays, Kreatech). Microarray hybridization was performed with 600 ng Cy3-labeled RNA for 17 h at 65 °C. Microarray raw data were obtained in an Agilent microarray Scanner C, Modell G2505C using Agilent Scan Control and Feature Extraction software version 10.7.3.1. and were processed using the limma R package (version 3.52.4)[69]. The data was normalized using the R-implementation in limma[69] and resulting *p* values from t-tests were corrected using the Benjamini and Hochberg[70] method to control the False Discovery Rate of differential expression. A |log₂ FC|≥ 1 threshold and a *p* value ≤ 0.01 were considered to indicate a significant change in gene expression. The full dataset is accessible in the GEO database with the accession number GSE183629 (Deletion and complementation of *alr3614* in *Anabaena* 7120).

## Northern blot analysis of mutants

Twenty micrograms of total RNA were separated on 10% polyacrylamide-8.3 M urea gels and electroblotted onto Hybond N nylon+ membranes (Amersham) with 1 mA per cm² for 1 h. CRISPR-related transcript accumulation was analyzed by Northern hybridization using single-stranded radioactively labeled RNA probes transcribed in vitro from PCR-generated templates (see Supplementary Table 3 for primers). The radioactively labeled probes were generated using [α-³²P]-UTP and the Maxiscript T7 In vitro transcription kit (Thermo Fisher Scientific). Hybridizations were performed in 50% deionized formamide, 7% SDS, 250 mM NaCl and 120 mM Na₂HPO₄/NaH₂PO₄ pH 7.2 overnight at 62 °C. The membranes were washed in buffer 1 (2 × SSC (3 M NaCl, 0.3 M sodium citrate, pH 7.0), 1% SDS), buffer 2 (1 × SSC, 0.5% SDS) and buffer 3 (0.1 × SSC, 0.1% SDS) each for 10 min. All wash steps were performed at 57 °C. Signals were detected with a Laser Scanner Typhoon FLA 9500 (GE Healthcare). Signal intensities of *cas12k* mRNA (indicated by a black triangle in Fig. 4e) were determined using Quantity One software and normalized to the signal of the 5S rRNA. For statistical evaluation *p* values were calculated from three biological replicates using two-tailed Student's *t* test

assuming equal variances using the data analysis add-in for Microsoft Excel 2019 software.

## Western blot analysis

Cyanobacterial cell harvesting and protein extraction were performed following established protocols[71]. Total proteins extracted from the samples were separated by 15% SDS–PAGE according to the standard procedure and electroblotted onto PVDF membranes (Amersham). To check for equal loading, the membrane was stained with Ponceau S (0.1% (w/v) in 5% acetic acid). After destaining, the PVDF membranes were blocked in 3% skimmed milk-TBST (0.05% Tween-20 in TBS) at room temperature for 30 min. Then, the membranes were incubated with Anti-DYKDDDDK (FLAG tag) Mouse Monoclonal Antibody (#HT201, 1:3000, TransGen) for 1 h and washed three times with TBST (15 min each). After that, the membranes were incubated with a Goat Anti-Mouse IgG/AP (#K0031G, 1:3000, Solarbio) for 1 h and washed three times with TBST (15 min each). Finally, signals were detected with NBT (nitro-blue tetrazolium chloride) and BCIP (5-bromo-4-chloro-3′-indolyphosphate p-toluidine salt) (#PR1100, Solarbio). For Western blot analysis of TXTL samples, 1 μL (Fig. 5d) or 9 μL (Fig. 8c) of the TXTL reactions were loaded on a 15% PAA separating gel. Proteins were electroblotted onto nitrocellulose membranes with a pore size of 0.45 μm (Amersham). The Penta His HRP conjugate kit (Qiagen) was used at a titer of 1:2000 to detect the 6xHis tag.

## Quantitative real-time PCR

Total RNA extraction, removal of the genomic DNA and reverse transcription were performed using a Bacteria RNA Extraction Kit and HiScript III RT Supermix for qPCR (+gDNA wiper) kit (Vazyme) according to the manufacturer's instructions. SYBR Premix ExTaqTM (Takara, Dalian, China) was used for qRT–PCR, and the cycle thresholds were determined using a Roche LightCycler® 480 II sequence detection system (Roche, Shanghai, China). *rnpB* (RNase P subunit B) was used as the internal control. The primers for *alr3614* (*cvkR*) and *rnpB* are listed in Supplementary Table 3. Three independent experiments were performed, which showed consistent results.

## TXTL assays

To test promoter fragments in a cell-free transcription-translation system, the *E. coli*-based TXTL assay was used[37,72]. The myTXTL Sigma 70 Cell-Free Master Mix was purchased from Arbor Biosciences. The included p70a plasmid was used as a template for cloning of the promoter sequences P*tracr*, P*cvkR*, P*cas12k* and P*psbAI* (Fig. 5b) in an open reading frame with the destabilized enhanced GFP (deGFP) and its 5′UTR. All PCRs were performed using PCRBio HiFi polymerase (PCR Biosystems). Promoter sequences were PCR-amplified from genomic DNA of *Anabaena* 7120 with overlaps to p70a. The p70a plasmid was also PCR-amplified.

The CvkR protein (Alr3614) and mutated variants with potentially reduced DNA-binding affinity (R16A-R20A-Q23A-K40A-R42A-N66A; CvkRmut and R42E) were used for analysis in the TXTL assay. The *cvkR* sequence was PCR-amplified from genomic DNA of *Anabaena* 7120. The corresponding DNA fragment for CvkRmut was ordered from IDT as gBlocks and subcloned into pJet1.2/blunt (Thermo Fisher Scientific) and then amplified via PCR with overhangs to pET28a. The plasmid pET28a was PCR-amplified as well. The CvkR and CvkRmut proteins were thus expressed from an IPTG-inducible T7 promoter with an N-terminal 6xHis tag and a TEV site for potential cleavage of the tag.

Fragment assembly via AQUA cloning was performed at room temperature for 30 min upon transformation into chemically competent *E. coli* DH5α cells for cloning. Assembled plasmids were isolated, and regions of interest were sequenced (Eurofins Genomics).

For the expression of proteins encoded on pET28a in the TXTL assay, T7 RNA polymerase (RNAP), expressed in this instance from p70a, and IPTG are necessary. Reactions were performed in duplicate

overnight at 29 °C in a total volume of 5 μL with 3.75 μL of TXTL master mix, 1 mM IPTG, 0.5 nM p70a_T7_RNAP, 1 pM pET28a_CvkR (or mutants: CvkRmut, R42E, S134C) and 5 nM p70a_promoter_deGFP. The fluorescence of deGFP was measured every 10 min (excitation 485 nm, emission filter 535 nm) in a plate reader (Wallac 1420 Victor2 micro-plate reader from Perkin Elmer).

## Protein heterologous expression and purification

For heterologous expression of CvkR in *E. coli*, the protein-coding sequence of CvkR was PCR-amplified from *Anabaena* 7120 genomic DNA and cloned into the pET28a-SUMO vector by using BamHI and XhoI restriction sites to generate the pET28a-SUMO_CvkR expression plasmid, which expresses CvkR with an Ulp1-cleavable N-terminal 6xHis-SUMO fusion tag. The plasmids pET28a-SUMO_CvkR-R42E and -S134C used for generation of R42E and S134C mutant proteins were constructed by point mutation method using plasmid pET28a-SUMO_CvkR as a template. The plasmid pET28a-SUMO_CvkRmut used for generation of CvkRmut (R16A-R20A-Q23A-K40A-R42A-N66A) was constructed by seamless assembly using pET28a-SUMO_CvkR as backbone and the ordered DNA fragment for CvkRmut in the TXTL assay as insert cassette. The sequence-verified plasmids were then transformed into *E. coli* BL21(DE3) for protein expression.

The expression strains were grown in LB medium to approximately OD$_{600}$ ~ 0.6 at 37 °C and induced at 25 °C overnight with 0.4 mM IPTG (isopropyl β-D-thiogalactopyranoside). After induction, cells were harvested by centrifugation and lysed in lysis buffer A (20 mM Tris-HCl, pH 8.0, 300 mM NaCl, 10 mM imidazole, 2 mM β-mercaptoethanol, 20 μg mL$^{-1}$ DNase I) using high-pressure homogenizer or ultrasonic cell disruptor. After centrifugation at 20,000 × $g$ for 50 min, the supernatants were loaded onto a Ni Sepharose 6 FF column (GE Healthcare), washed with lysis buffer A containing 40 mM imidazole, and eluted with 20 mM Tris-HCl (pH 8.8), 300 mM imidazole, and 2 mM β-mercaptoethanol. The eluted fractions were treated with Ulp1 protease overnight at 4 °C and purified with a HiTrap Heparin HP column (GE Healthcare) to remove the 6xHis-SUMO fusion tag and impurities. Subsequently, tag-removed target protein fractions were further purified by Superdex 75 10/300 GL (GE Healthcare) with 10 mM Tris-HCl (pH 8.8), 100 mM NaCl, and 2 mM β-mercaptoethanol. Finally, target proteins were collected, concentrated and stored in the same buffer. All mutants were purified following the same protocol used for preparation of wild-type CvkR protein.

Selenomethionine-labeled (Se-Met) CvkR protein was over-expressed and purified using the same procedures described above; however, the medium was substituted with M9 medium, and seven essential amino acids were added at the mid-log phase before induction. To prevent oxidation of the selenium atoms, 5 mM β-mercaptoethanol was added to the final elution fraction containing Se-Met CvkR.

## Analytical gel filtration

Size exclusion chromatography was performed to probe the molecular weight of CvkR and the relevant mutant proteins in solution using Superdex 75 10/300 GL column connected to an ÄKTA Pure system (GE Healthcare). The column was calibrated with a gel filtration calibration kit LMW (Low Molecular Weight) (GE Healthcare) in a buffer containing 10 mM Tris-HCl (pH 8.8), 100 mM NaCl, 2 mM β-mercaptoethanol. About 500 μL proteins (~0.6 mg mL$^{-1}$) were loaded onto to the system using a 500 μL loop. Calibration curves were used to calculate the oligomeric state of untagged CvkR, R42E, S134C and CvkRmut according to their elution volumes. The calibration curve based on the molecular markers (Cytiva) (Conalbumin:75 kDa, Ovalbumin: 44 kDa, Carbonic anhydrase: 29 kDa, Ribonuclease A: 13.7 kDa, Aprotinin: 6.5 kDa) is lgMr = −1.7621Ve + 4.8613 ($R^2$ = 0.9841, Ve: elution volume; Mr: molecular weight).

## Dynamic light scattering analysis

DLS analysis were performed using a DynaPro99 DLS plate reader instrument (Wyatt) equipped with an 830-nm laser source. CvkR protein was used at a final concentration of 50 μM in a buffer containing 10 mM Tris-HCl (pH 8.8), 100 mM NaCl, and the measurement was carried out for 10 min at 25 °C. The Wyatt Dynamics software (version 7.0) was used to schedule data acquisition and calculate the hydrodynamic radius.

## EMSA assays

The double-stranded DNA probes for EMSA were acquired by annealing the 5′-biotin-labeled oligonucleotides and their complementary strands (Tsingke Biotechnology Co., Ltd.) (Supplementary Table 3) in equimolar concentrations (10 μM) with a thermocycler (94 °C for 5 min, then with a temperature reduction of 1 °C per min to 25 °C). The EMSA assays were performed using a chemiluminescent EMSA kit (Beyotime, China) according to the manufacturer's protocol. Each double-stranded DNA probe with a concentration of 2 nM and various amounts of CvkR or mutant proteins of interest were mixed with the EMSA binding buffer to a final volume of 10 μL and incubated at 25 °C for 30 min. The binding buffer contains 10 mM Tris-HCl (pH 8.0), 25 mM MgCl$_2$, 50 mM NaCl, 1 mM dithiothreitol, 1 mM EDTA, 50 μg mL$^{-1}$ poly(dI-dC), 0.01% Nonidet P-40, and 10% glycerol. After binding, the samples were electrophoresed on a 6% native polyacrylamide gel with 0.5x Tris-borate-EDTA (TBE) buffer, followed by electrotransfer to a nylon membrane and crosslinking by a hand-held UV lamp. The bands were detected by using the BeyoECL Plus kit (Beyotime).

## DNase I footprinting assays

DNase I footprinting assays were carried out similar to previous research[73]. Specifically, ~300 nt FAM-labeled probes, which cover the promoter regions of *cas12k* or *tnsB*, were cloned into pUCm-T vector and PCR amplified with 2× TOLO HIFI DNA polymerase premix (TOLO Biotech, Shanghai) using primer pairs P3613-F(FAM)/P3613-R or M13F(FAM)/M13R (Supplementary Table 3), and purified by the Wizard® SV Gel and PCR Clean-Up System (Promega, USA). Binding reactions were performed in a total volume of 40 μL containing 50 mM Tris-HCl, pH 8.0, 100 mM KCl, 2.5 mM MgCl$_2$, 0.2 mM DTT, 10% glycerol, 2 μg salmon sperm DNA, 300 ng probes and 0.1 μg CvkR protein at room temperature for 30 min. Following DNase I treatment (Promega), phenol/chloroform extraction, and ethanol precipitation, products were dissolved in 30 μL MilliQ water. The preparation of the DNA ladder, electrophoresis and data analysis followed the protocol by ref. [73], except that the GeneScan-LIZ600 size standard (Applied Biosystems) was used.

## Phylogenetic analysis of CvkR and Cas12k

The identified CvkR and Cas12k proteins were compared to each other and analyzed for alternative start positions to correct potential incorrect annotations, as in the case of Alr3614 (BAB75313.1) and All3613 (BAB75312.1). Elongated N-terminal regions with no similarity to homologs encoded by other *cvkR* and *cas12k* genes were removed from the analysis. The sequences were then aligned by M-coffee[74,75] and further analyzed by the BEAST algorithm(v 1.10.4)[76]. The phylogenetic analyses were calculated with a strict clock model and a birth-death speciation process (Yule) as tree prior, using the standard parameters (clock.rate: fixed value = 1; treeModel.rootHeight: Using Tree Prior [0, infinity]; and yule.birthRate: LogNormal [1, 1.5], initial = 2) as well as the substitution model Blosum62. The MCMC chain was sampled for CvkR every 1000 steps over 1 million generations[77–79] and every 1000 steps over 10 million generations[77–79] for the analysis of Cas12k. In order to assure a reasonable effective sample size (ESS; threshold at 200) the results were analyzed by the Tracer v 1.7.2 tool[80] and the first 10% of

trees were discarded from the analysis. The trees were then generated by the tree annotator tool (from the BEAST package).

### Crystallization, data collection, and structure determination of the CvkR protein

Using a hanging drop-vapor diffusion method, both native and selenomethionine-substituted (Se-Met) CvkR crystals appeared at 18 °C in crystallization reagent containing 0.1 M phosphate citrate (pH 4.4-4.6) with 15%-20% PEG300. However, diffraction-quality crystals were obtained only when 10 mM ATP was added into the crystal screen droplet, which consisted of a 1:1 (v/v) protein at 15 mg mL$^{-1}$ and the well crystallization reagent. Before flash cooling in liquid nitrogen, crystals were cryo-preserved using crystallization reagent supplemented with 15% PEG400.

The diffraction data were collected at wavelengths of 0.97930 Å or 0.97918 Å on the Shanghai Synchrotron Radiation Facility, beamlines BL17U1 and BL18U1, in a 100 K nitrogen stream. Data indexing, integration, and scaling for native CvkR were conducted using HKL3000 software (version721.3)[81]. SAD X-ray diffraction data were processed by Aquarium pipeline[82]. Eleven selenium sites were located by HySS as implemented in AutoSol/Phenix (version 1.17.1-3660). The resulting main chain structure was used as the initial search model for molecular replacement by Phaser/Phenix (version 1.17.1-3660) to determine the native CvkR structure[83]. Structure refinements were iteratively performed using the programs Phenix and Coot (version 0.9.8.1 EL)[84,85]. The statistics for data processing and structure refinement are shown in Supplementary Table 4. The coordinates were deposited in the Protein Data Bank with the PDB ID 7XN2. Figures were prepared using PyMOL (version 2.4.0)[86].

### Reporting summary

Further information on research design is available in the Nature Portfolio Reporting Summary linked to this article.

## Data availability

Source data are provided as a Source Data file. The full transcriptome datasets for the WT and mutants Δ*cvkR* and Δ*cvkR*Com are accessible from the GEO database with the accession number GSE183629 (Deletion and complementation of *alr3614* in *Anabaena* 7120). The structural data can be accessed at the Protein Data Bank under the PDB accession number 7XN2. Other PDB structures referred to in this study are 1Q07 (CueR structure), 4R4E (GlnR-DNA structure), and 5D8C (HiNmlR-DNA structure). AlphaFold2 predicted McdR (NCBI: ABK74795.1) [https://alphafold.ebi.ac.uk/entry/A0QYF9]. Source data are provided with this paper.

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

## Acknowledgements

Financial support for this work was provided by the National Key Research and Development Program of China (Grant number 2021YFA0909700), the Joint Sino-German Research Program (grant HE 2544/13-1 to WRH and grant M-0214 to X.L.), the DFG (grant HE 2544/14-2 to W.R.H.), the National Natural Science Foundation of China (Grant numbers 31525002, 31570068 and 31761133008), the QIBEBT and Dalian National Laboratory for Clean Energy (DNL), CAS (grant QIBEBT I201904 to T.Z.), and the Shandong Taishan Scholarship

(to X.L.). We thank the staff from BL17U1 and BL18U1 beamlines at Shanghai Synchrotron Radiation Facility for assistance during data collection.

## Author contributions

W.R.H., X.L., and T.Z. designed the study. T.Z., Y.X., and H.L. constructed the *Anabaena* 7120 mutant strains. V.R. performed the TXTL experiments, microarray analyses and Northern hybridizations. M.Z. did the majority of bioinformatic analyses. Y.S. and Y.X. performed the qRT–PCR and Western blot analyses. Y.S., T.Z., and Y.L. performed EMSA and DNase I footprinting experiments. Y.L. and H.M. performed the structural analysis of CvkR. V.R., M.Z., Y.L., T.Z., and W.R.H. wrote the paper with contributions from all authors.

## Funding

## Competing interests

The authors declare no competing interests.
