## [Peer Review File · Nature Communications]

CvkR is a MerR-type transcriptional repressor of class 2 type V-K CRISPR-associated transposase systemsREVIEWER COMMENTS

Reviewer #1 (Remarks to the Author):

The manuscript “CvkR, a novel MerR-type transcriptional regulator, is a repressor of class 2 type V-K CRISPR-associated transposase systems” by Ziemann and colleagues reports structural and experimental data on MerR-like repressor of type V-K CAST systems. This study sheds light on how these transposable elements are controlled. Authors found that deletion of CvkR leads to the overexpression of tracrRNA, crRNAs, Cas12k and Tn7-like transposase subunits. They also identified a binding site of this regulator and solve its high-resolution structure. While overall reported findings are of interest and most conclusions seem to be solid, there are some issues with bioinformatic analysis and some inconclusive data on ATP binding. Below I outline my concerns and suggestions:

1. My main concern is about phylogenetic analysis. The Figure 2 title is “Phylogenetic tree of all CvkR homologs”, however the tree contains non-homologous sequences of two different folds HTH and RHH, so the sequences cannot be meaningfully aligned and thus the entire analysis is simply incorrect. Phylogenetic analysis could be performed for homologous sequences only and only when there is enough (roughly >50) phylogenetically informative positions (the program BLOCKS can be used to identify such positions in a given alignment). Next, a purpose of such analysis should be clearly formulated. If this is, for example, a question on monophyly of type V-K CAST associated MerR regulators, then the set should also include all other MerR regulators present in these genomes and closest CvkR homologs from other genomes, otherwise authors cannot claim that these genes are monophyletic, because without such comparison they can group because they have the same fold. So, I suggest reconstructing a phylogenetic tree using Cas12k sequences (optionally or additionally TnsB or TnsC) and map the type of regulators (MerR, RHH and also specify other types of DNA-binding proteins) to the respective tree branches. This presentation then will answer a question if there is a shuffling of these regulators in these loci or they largely follow the evolution of Cas12k.
2. Please provide description of BLASTP search parameters which were used for identification of all components of CAST system and more details on how “small DNA-binding proteins” were identified.
3. Is there any reason to believe that a type V-K CAST regulator should be encoded next to Cas12k? If yes, what is that reason? And if no, these genes should be searched for elsewhere within each Tn7 locus and included in a supplementary phylogenetic analysis of respective families to check if they indeed co-evolve with components of the CAST system.
4. Please provide more details on what genes were differentially regulated (Fig. 5c), could some of them be an artifact? Is it possible to identify a CvkR binding site in the promoter regions of the genes outside CAST locus? If not it is desirable to perform DNase I footprinting assay for respective promoter regions (at least for selected up- and downregulates genes).
5. A large part of the paper discusses ATP binding interface, which authors believe is not a natural ligand. This makes this part of the paper inconclusive and vague. I suggest streamlining of this section.

6. Related to 1. It is misleading to call non-homologous proteins of HTH and RHH family by the same name CvkR. For simplicity I suggest to keep the name only for MerR-like repressors, but do not assign it to RHH and other putative type V-K CAST regulators.

Minor corrections:

1. Line 70-71. I-F CAST encodes for five core proteins: Cas6, Cas7, Cas5, Cas8; the latter two are fused.
2. Line 90. The logic behind this sentence is not clear: "Thus, the tight regulation of these systems can be expected." A reference to previous work/reviews would help.
3. Line 159. "The majority of cargo genes, located between cas12k and tniQ, are significantly more divergent". What is the meaning of "divergent" here?
4. Figure 6B. Why the upstream -35 (blue) segment of promoter in the Anabaena 7120 is colored differently compared with identical segments in the alignment of other species? Same question applied for -10 region of CvkR. Explain better in the legend how these regions in other genomes were identified and colored.

Reviewer #2 (Remarks to the Author):

The authors have identified a new MerR-family transcription regulator and proven/characterized the role of this protein as a repressor of class 2 type V-K CRISPR-associated transposase systems by extensive in vivo and in vitro experiments. In addition, a 1.5 Å crystal structure of the homodimer in complex with ATP was determined to provide structural insights, which is highly similar to other MerR family members except for some discrepancies in dimerization and effector binding domains. In general, although this study is not the first to discover/predict the existence of MerR-family regulators in CAST systems (see ref 14), it did provide solid data to identify and characterize the repressor role of this protein and the results will enrich the understanding of MerR-family transcription regulators. However, considering MerR-family members have been well and extensively studied and no novel mechanistic information on how MerR factors regulate transcription has been supplemented by this study, this reviewer has not been convinced that it is qualified for publication in Nature Communications and would suggest submitting it to a more specific journal.

Specific comments:

(1) It is one of the common phenotypes that a transcriptional regulator would regulate the transcription of nearby genes. Therefore, it is not surprising that CvkR would regulate the transcription of CRISPR system. The mechanism of this regulation is one of the most important questions. However, the authors have not shown a clear answer on how CvkR regulates this CRISPR system. For example, the signal triggering the conformational change of CvkR has not been demonstrated. The structures of CvkR with

and without effector have not been compared. How does CvkR bind to promoter and influence the transcription carried out by RNA polymerase has not been investigated?

(2) A structure of CvkR in complex with promoter DNA is needed to support the relevant presentation (lines 45-48, 411-428) in the manuscript. In the meantime, additional superimpositions with CueR repressor complex or other MerR-DNA repressor complex should be included, which will provide support for the description on the putative interaction between CvkR and promoter DNA. More specifically, lines 413-415: It's impossible to predict R42 is likely to participate in the binding of DNA based on Figure 9C. A figure of superimposing the structure of the previous MerR-DNA complex is necessary. The further description (lines 418-428) also needs relevant figure to support it.

(3) About the structural model shown in the PDB validation report:

(i) Rfree value (23.1 %) is slightly low considering reporting a 1.5 Å X-ray crystal structure.

(ii) RSRZ outliers (20.9 %) are too high. The real-space R-value (RSR) is a measure of the quality of fit between a part of an atomic model and the data in real space.

These suggest that the deposited pdb model needs to be improved.

(4) Figures of SAD experimental map and final 2fofc map should be provided to show the quality.

(5) In Figure 8, it's better to present superimpositions using homodimer structures to clearly show the distinct dimerization and effector binding domains in this protein. The current figure does not project these discrepancies.

(6) ATP binding figure (Fig 9) is poorly presented. It's better to hide hydrogen when making figure and a clearer presentation is needed.

(7) Authors obtained a structure of CvkR homodimer in complex with ATP, which binds at the putative effector binding region. Although the authors suggest cyclic oligonucleotide family molecules are the effector of CvkR, no candidates have been identified. This reviewer is curious what attempts have been tried. The possibility of cyclic oligonucleotide or even ATP as an effector could be easily tested. A control analysis with/without cyclic oligonucleotide/ATP in the in vitro DNA binding experiments (EMSA) should be considered and discussed. The reported structure may not represent the repressor state. Please note that the available structures of CueR homodimer have three different states/conformations: CueR dimer with effector Ag/Cu(I) (pdb: 1q07), CueR-DNA repressor complex (pdb: 4wls), and CueR-DNA activator complex (pdb: 4wlw).

(8) Line358-361: Unit of contact area should use Å². Contact area analysis in this study was calculated using the interface between one CvkR molecule in an asymmetric unit and its symmetric mate. Are there other possible dimerization interfaces found in the crystal packing? If so, what are the other interface areas? Smaller than 970.1 Å or not? What are the contact areas of dimerization in other MerR factors? In addition, this contact area can only suggest stronger/weaker interaction between two CvkR molecules. The dimerization state of this protein in solution should be determined by other methods, such as SEC and/or light scattering analysis. The relevant sentences need to be modified.

(9) A superdex 200 10/300 column used in the study is not a good one to determine the dimeric or monomeric state of CvkR (~17 kDa). A superdex 75 10/300 column and light scattering analysis should be utilized.

(10) Line 429 6 aa were selected to be mutated to generate the CvkRmut to study the relevance of DNA binding and transcriptional regulation. It will be good to first analyze its oligomeric state in solution using SEC and then conduct a DNA binding ability assay (EMSA) before the final TXTL functional assays.

(11) Mutations have been constructed in the putative DNA binding domain. Since the dimerization domain is another distinct region of its structure, this reviewer suggests a relevant mutagenesis study on this region to strengthen the relevant statements.

(12) The location of CvkR binding region in the target promoter is important for explaining the regulatory mechanism. Authors may present the CvkR binding region and promoter -10 and -35 elements in a main figure.

(13) The classical MerR family regulators (such as CueR, BmrR) have been demonstrated as activator, but some proteins have been shown as repressor (McdR, HonC). Authors may discuss the potential mechanism of CvkR by comparing with these regulators.

(14) Authors demonstrated that CvkR could be translated from leaderless mRNA, but did not show the meaning of this leaderless mRNA translation. It might be interesting to test if the functions of CvkR-L and CvkR-S are different.

Reviewer #3 (Remarks to the Author):

In the manuscript, the authors report a transcription regulation mechanism for repressing the basal activity of CRISPR-associated Transposons (CASTs). The transcription repressor, CvkR of CAST in *Anabaena* sp. PCC 7120 was chosen to study the detailed repression mechanism. The study shows that CvkR represses the expression of cas12k as well as other genes encoding essential components of CAST. A long DNA-recognition motif of CvkR was identified in the core promoter region of cas12k and a high-resolution crystal structure of CvkR is reported. Overall, the manuscript explains how bacterial cells restrict the basal activity of CAST by showing that CAST is transcriptionally repressed by a new MerR-type transcription factor. However, more evidence should be collected to demonstrate the specific binding of CvkR to its predicted binding motif and more structural data, ideally a crystal structure of CvkR-DNA binary complex, should be provided to explain how CvkR recognizes the long DNA binding motif. The detailed comments are listed below.

1. Please add figure citations in the second and third paragraph on Page 6 and the first paragraph on page 7.
2. Figure 2. The authors reported that at least one regulator near *cas12k* gene was identified in 94 CASTs. Please clarify how many CASTs were surveyed.
3. Fig 3B does not add convincing argument for the shorter ORF of *CvkR-S*.
4. The distance between *CvkR* and *Cas12k* genes are less than 100 nt (Fig. 3B), does the binding of *CvkR* on its predicted cis element affects the expression of *cvkR* itself?
5. Fig. S1C. Please add the explanation for the yellow and red bars in the figure legends.
6. Fig. 3c. Please explain the WT. I suppose it means the *cvkR* depletion strain but not the genuine 'wild-type' strain.
7. Fig. 3D shows *CvkR-L* can be translated under control of an artificial promoter, thereby did not provide support for the only existence of *CvkR-S* in bacterial cells. The author states that 'the start codon of *alr3614S* coincides with the previously mapped TTS of its mRNA' (line 204). This is a better argument and deserves a supplemental figure showing the mapped TTS, start codon, the predicted -35/-10 elements of the *cvkR* promoter.
8. Fig 4B-E. Please label the positions of the target bands. In Fig. 4B, I did not see much difference between the wt and *cvkR* depletion strains. Does that mean the expression of *cvkR* gene is repressed in wt bacterial cells? In Fig. 4D, What is the identity of the bands of ~ 500 nt that showed strong signals in wt bacterial cells but were absent in *cvkR* depletion and *cvkR* complementation strains. Fig S2 suggests that the *tracrRNA*-CRISPR array has its own promoter. How would it explain *cvkR* depletion also increases the transcription level of *tracrRNA* and CRISPR? Is there a *CvkR* degenerate binding motif of *CvkR* on that promoter? Did the authors test whether *CvkR* interacts with the promoter in a sequence-specific manner?
9. Fig. 5C show *TnsB* is also upregulated upon depletion of *cvkR*. Is there a degenerate *CvkR* binding motif on *TnsB* promoter? Did the authors test whether *CvkR* interact with the promoter in a sequence-specific manner?
10. Please indicate the TSS in Fig. S2A and S2B
11. Please label the target *cvkR* band in Fig. 5A.
12. In Fig. 7A, the result is ambiguous. The presence of 5 nM *CvkR* shifted majority of DNA in all tested promoters, while the presence of lower concentration of 1.5 or 1 nM *Cvkr* shifted DNA in a similar extent in all tested promoters. Moreover, it is better to include point mutations (or combination of point mutations) of the *CvkR* motif in the EMSA and deGFP reporter assays to validate the binding motif.
13. In Fig. 7F, 5 nM *CvkR* is used to repress *ptracrRNA* instead of 1nM *CvkR* as used in Fig. 7D and 7E. Does higher *CvkR* protein level cause non-specific DNA interaction and thus non-specific transcription repression? A control should be included to prove that neither 1nM nor 5nM *CvkR* repress transcription from a non-relevant promoter.
14. Line 359, '970.1 Å' should be '970.1 Å²'

15. Fig. S3, please add the calculated molecular weight in the figure legend.
16. The two half palindromic sites of the CvkR motif (5'-AAAACACA-N21-TGTGTTTT-3') is separated by a 21-bp spacer, which is much larger than the spacer length of typical MerR-family TF motifs, and much larger than the spacer length of binding motifs of other bacterial TFs. To recognize the two half palindromic sites spanned by 29 bp (21+4+4) simultaneously by the CvkR homodimer, The DBDs of the two CvkR protomers should be separated by ~ 100 Å, much larger than the current distance of the two DBDs in the crystal structure (28 Å; Fig. 8B). Therefore, the CvkR dimer must undergo drastic conformational change upon DNA interaction. It is unknown whether and how the CvkR dimer is fully stretched to bind its long motif. A crystal structure of DNA-bound CvkR would explain how CvkR interacts with the long motif.
17. Line 382 and Figure 8c, please add citations for the CueR and SoxR structures used for comparison.
18. Line 393, 'Notably, the hydrophobic residue W133, originally embedded in the hydrophobic interior, is exposed to the solution side due to the binding of ATP'. The authors compared the W133 conformation in the presence or absence of ATP binding. Which structure was used to represent the W133 conformation in the absence of ligand?
19. Lines 416-417, It is better to delete the sentence. Residues making interactions with ATP does not necessarily indicate its capacity of interaction with DNA.
20. Line 418-428, please prepare a supplementary figure to show the structure superimposition.
21. Fig 10C. It is surprising that a combination of six point mutations didn't abolish the sequence-specific DNA recognition by CvkR (5nM). Is it because other potential key DNA-contacting residues are not included in the point mutations?
22. Line 484, recent structural works of transcription activation complexes the MerR-TFs should be cited.

Reviewer #4 (Remarks to the Author):

This manuscript by Ziemann and colleagues describes a series of phylogenetic, genomic, cellular, structural and molecular studies on CvkR from the cyanobacterium, *Anabaena* sp. PCC 7120. CvkR is the authors suggestion and is very reasonable. Their data reveal that CvkR is a repressor of the class 2 type V-K CRISPR-associated transposase system. Moreover, their structural work reveals a new subfamily of the MerR superfamily. This study is very interesting and for the greater part the work is well done. However, there are several issues that the authors should address.

The authors refer to the "novel" C-terminal domain, with a standard alpha helix ($\alpha 7$) and lacking the cysteine that is conserved in the other CvkR family members, which the authors identify in their

phylogenetic studies. I agree that the C-terminal domain and the dimerisation mechanism is new, but the importance of helix $\alpha 7$ is unclear with respect to effector binding. The authors should change the serine found in their CvkR protein to a cysteine to assess its importance. From reading the manuscript, the authors imply this change is important, but provide no evidence to support this implication. Also, the authors should consider removing $\alpha 7$ and test the functionality of the resulting CvkR $\Delta 7$.

A larger issue that the authors must address is the inclusion and discussion of other MerR families. They completely ignore TnrA and GlnR, which form a different branch of the MerR family. Indeed, these MerR family members have very distinct C-termini and N-termini that are involved in dimerization. Further, these proteins bind protein to effect their transcription regulation. They also do not bind the canonical MerR DNA binding site, i.e., one in which the -10 and -35 boxes are separated by at least 18 base pairs. The authors also fail to include any discussion of BldC, which is a critical regulator involved in development and oligomerises and binds DNA in a fashion different to MerR, BmrR, CueR etc. and likely CvkR. A more in-depth discussion of these other MerR proteins is necessary.

The authors present a DNA binding site but do not measure the affinity of CvkR for this site. From the presented EMSA experiments, the affinity would not appear to be that high. They should determine the affinity and follow this up with mutation of the palindrome and the spacer, both sequence and length.

The crystal structure was obtained to high resolution but required the presence of ATP in the crystallisation drop. On page 16 the authors describe the ATP binding site and the interactions. This is not well done. As the specificity for the adenine base is not described fully. From Figure 9, there is no way to tell how the N6 (hydrogen bond donor) and N1 (hydrogen bond acceptor) of the ring are "read" by the protein. It looks like the peptide backbone is involved. Further, the pi-pi and cation-pi interactions contribute to affinity but not to specificity. The authors should mutate at least residue R136 to understand its importance in ATP binding (see below). Moreover, the authors write, "Notably, the hydrophobic residue W133, originally embedded in the hydrophobic interior, is exposed to the solution side due to the binding of ATP...". First, W133 is aromatic, not hydrophobic, and as such this residue type can be found on the surface of a protein. More, how do the authors know that W133 is "originally embedded in the hydrophobic interior"?

The authors should measure the affinity of the ATP for CvkR as well as the affinities of ADP and AMP. They should also measure the affinities of CTP, which could present N4 (hydrogen bond donor) and N3 (hydrogen bond acceptor) to the protein, and GTP, which should not be compatible because it would present O6 (hydrogen bond acceptor) and N1 (hydrogen bond donor).

The authors identify a palindrome-containing DNA binding site via DNase I protection. There are four adenines on one end and four on the other end. Although these A tracts are possibly part of the CvkR

cognate site, DNase I does not do a very good job in cutting these types of sequences. The suggestion here would be just to couch the wording that the A tracts are not always cut by DNase I.

Page 19: Again, in the Discussion there is nothing about GlnR/TnrA or BldC. Also, the authors should include the name(s) of “MerR-like proteins without effector interactions...”. MtaN is probably also in this category.

Page 21: “the efficient binding of an adenine points to a metabolite that may be related to the cyclic nucleotide family of signalling molecules...”. What does efficient mean here? You have added 10 mM ATP as a crystallisation reagent. Without a K_d, this is not an appropriate statement.

Page 27, line 674: “...0.3 mg CvkR...”. Please provide the concentration, not the amount.

Page 27, line 678: The names and masses of each molecular weight marker should be given here and included in the Supplementary Figure S3.

Figure 2 suggestion: It would be helpful if the authors could label the top of the columns with something like RT, DBD, O. This would make it easier for the reader to relate the data to the Repressor Type, DNA-binding Domain and Organism.

Figure 8C: The authors do not really provide much of a discussion concerning the overlays that are presented. Furthermore, they did not include other MerR family members in the overlays including TnrA/GlnR, BldC or BmrR.

Methods: In several instances the authors refer to previously published work for a description of the methods that they employ in this work. They should include more information as this is a bit of a disservice and inconvenience to the reader.

Minor:

Page 14, line 347: Suggested word change: “The results shown in Figs. 4 to 7...” might be better stated as “Our data has established...”

Page 16, line 389: “binds exactly to the putative effector-binding domain...”. What does exactly mean here? This is not likely the best choice of adverb.

Page 19, line 467: The authors should refer to the original MerR papers by Walsh and Summers, not just a reference to two general reviews.

Page 19, line 478: "...so MerR represses the binding of the σ factor." This is not quite correct. The typical repressive MerR proteins block σ factor binding by altering the DNA conformation around the -10 and -35 promoter elements. The work repress implies something else.

Page 29, line 717: "...15%-20% PEG300." Do the authors mean PEG3000?

Figure 4: The authors should point out the differences in the bp ladders between B-D and E.

Point-to-point replies to the reviews of our manuscript NCOMMS-22-21094-A

REVIEWER COMMENTS

Reviewer #1 (Remarks to the Author):

The manuscript “CvkR, a novel MerR-type transcriptional regulator, is a repressor of class 2 type V-K CRISPR-associated transposase systems” by Ziemann and colleagues reports structural and experimental data on MerR-like repressor of type V-K CAST systems. This study sheds light on how these transposable elements are controlled. Authors found that deletion of CvkR leads to the overexpression of tracrRNA, crRNAs, Cas12k and Tn7-like transposase subunits. They also identified a binding site of this regulator and solve its high-resolution structure. While overall reported findings are of interest and most conclusions seem to be solid, there are some issues with bioinformatic analysis and some inconclusive data on ATP binding. Below I outline my concerns and suggestions:

We appreciate the critical but productive comments. We have performed the suggested additional analyses and added the results to the manuscript as detailed below.

1. My main concern is about phylogenetic analysis. The Figure 2 title is “Phylogenetic tree of all CvkR homologs”, however the tree contains non-homologous sequences of two different folds HTH and RHH, so the sequences cannot be meaningfully aligned and thus the entire analysis is simply incorrect. Phylogenetic analysis could be performed for homologous sequences only and only when there is enough (roughly >50) phylogenetically informative positions (the program BLOCKS can be used to identify such positions in a given alignment). Next, a purpose of such analysis should be clearly formulated. If this is, for example, a question on monophyly of type V-K CAST associated MerR regulators, then the set should also include all other MerR regulators present in these genomes and closest CvkR homologs from other genomes, otherwise authors cannot claim that these genes are monophyletic, because without such comparison they can group because they have the same fold. So, I suggest reconstructing a phylogenetic tree using Cas12k sequences (optionally or additionally TnsB or TnsC) and map the type of regulators (MerR, RHH and also specify other types of DNA-binding proteins) to the respective tree branches. This presentation then will answer a question if there is a shuffling of these regulators in these loci or they largely follow the evolution of Cas12k.

Many thanks, we are really grateful for the suggestions! We wanted to visualize the association of different regulators with the existing CAST systems.

Following the advice by reviewer #1, we have replaced the previous Fig. 2 with a tree showing the phylogenetic analysis of Cas12k variants and then plotted their associated regulator types alongside (now Fig. 2A). We excluded CAST systems with degenerated *cas12k* genes yielding 106 Cas12k proteins and 292 phylogenetically informative positions.

This analysis revealed that the three different groups of regulators each are associated with distinct branches of Cas12k homologs, indicating three distinct association events.

Previous Fig. 2 has been replaced and corresponding sections in the text and in the figure legend have been adapted accordingly.

2. Please provide description of BLASTP search parameters which were used for identification of all components of CAST system and more details on how “small DNA-binding proteins” were identified.

We now provide all search parameters.

Regarding the details on our search for “small DNA-binding proteins”: After identifying the 118 CAST systems and determining their left and right ends, we set up a library of proteins from all ORFs inside the CAST transposons requiring a minimum length of 50 AA and allowing ATG, TTG or GTG as possible start codons. In this database, we searched for homologs to the CvkR regulator of *Anabaena* 7120 (Alr3614) (maximum e-value of 1e-20) yielding 53 *merR*-like genes, all of which were located upstream of *cas12k* in reverse orientation, like *alr3614*.

Next, we used the CD-search webtool of NCBI¹ to identify known domains in the proteins in our library. We looked specifically at genes upstream of *cas12k* in reverse orientation and found 10 genes with a cl10310-domain (PHA01623 superfamily) of unknown function (later named “Arc_1”) and 6 genes with cl06769-domain (RHH_5, ribbon-helix-helix; later named “Arc_2”). A further analysis with HHpred² identified both sets of genes to encode small (53-72 AA) Arc repressors, without pronounced further sequence similarities. In order to identify other variants, we searched for homologs of these Arc repressors within the CAST systems (maximum e-value of 1e-20) and found a total of 24 Arc_1 and 11 Arc_2 genes, which were all located upstream of *cas12k* in reverse orientation, except for two Arc_1 homologs. One of these other Arc_1 genes is with a distance of 1.3 kb further away from the *cas12k* locus, while the other one seems to have no association to *cas12k*.

We further investigated this significance by searching for additional homologs to these regulators outside of the known CAST systems and identified 210 unique genes (157 CvkR-like, 32 Arc_1-like and 21 Arc_2-like; maximum e-value of 1e-20). We searched for *cas12k* in close vicinity to these genes and found that in 169 cases a *cas12k* gene or a degenerated version of *cas12k* was present (see our new Table S2 in the manuscript). We also searched for other CAST components around the regulators and found 130 left end elements, 90 CRISPR arrays, 119 tracrRNA loci and 139 tRNA genes. Just in 18 cases we could not find any CAST components close to the regulator.

This high association with the CAST system supports the importance of these regulators for the transposon and *cas12k*.

Thus, there are three main CAST-associated regulator types. The remaining unclassified proteins with an HTH domain showed less association with *cas12k* making speculations about the possible biological role of those proteins ambiguous.

We added the corresponding information in the text (section “Three types of regulators are associated with the CAST systems of cyanobacteria”, replaced Fig. 2 and provide the new Supplemental Tables S1 and S2.

3. Is there any reason to believe that a type V-K CAST regulator should be encoded next to *Cas12k*? If yes, what is that reason? And if no, these genes should be searched for elsewhere within each Tn7 locus and included in a supplementary phylogenetic analysis of respective families to check if they indeed co-evolve with components of the CAST system.

Yes. The CAST V-K system is a mobile and frequently changing genetic element, indicated by the high variation in the number and composition of cargo genes. We could show that the regulators are commonly associated with *Cas12k* or other CAST-components (see also our reply to point 2 above); therefore, we assumed a functional connection between the regulators and the CAST system.

In our previous analysis (Hou, et al., 2019)³ we had identified multiple examples of a *merR*-like gene suspiciously close to a *cas12k* gene. The distance between both genes is indeed very low (usually <100 nt), and, as we show in the current analysis for *Anabaena* 7120, with overlapping promoters and the expression of both proteins from leaderless mRNAs.

We have performed the suggested additional analyses (Fig. 2A and 2B) and added the results to the manuscript.

4. Please provide more details on what genes were differentially regulated (Fig. 5c), could some of them be an artifact? Is it possible to identify a *CvkR* binding site in the promoter regions of the genes outside CAST locus? If not it is desirable to perform DNase I footprinting assay for respective promoter regions (at least for selected up- and downregulates genes).

Detailed information on differentially regulated genes was given in Table S3. To answer the 2nd point, we now write on p12/13: “Differentially expressed genes outside the AnCAST system included the L-array, a cryptic tRNA gene cluster relevant for survival under translational stress⁴, which was upregulated here in $\Delta cvkRCom$ (Fig. 4G), inverse to the regulation observed for *cas12k* and *tnsB*. Other differentially regulated genes were the L-array adjacent gene *all8564* encoding an HNH-type homing endonuclease, *rtcB* encoding an RNA ligase associated with RNA

*repair*⁴ as well as *alr0739* and *alr0740* encoding a Ydel and a slipin family homolog. The effects on these genes could have been caused by the presence of erythromycin to stabilize the plasmid introduced in $\Delta cvkR$ Com. Therefore, the up- and downregulated genes outside of AnCAST cannot be safely associated to the CvkR regulon.”

Therefore, as proposed by this reviewer, we focused on the improved characterization of the CvkR binding site by performing additional EMSA assays using probes of P_{cas12k} with various lengths and sequence mutations during the revision of the manuscript. We inferred from these analyses that the previously proposed CvkR binding site was too long. To specify the recognition motif of CvkR further, we performed additional DNase I footprinting assays with the promoters of *tnsB* (P_{tnsB}) and *tracrRNA* (P_{tracr}). We finally confirmed that CvkR can directly bind to P_{cas12k} and P_{tnsB} , but not P_{tracr} , and identified a 15 nt CvkR recognition motif, 5'-AnnACATnATGTnnT-3'. This was further verified by EMSA assays with mutations of the palindrome and the spacer, both in sequence and length, as also suggested by Reviewer #3. We added the corresponding information in the text (section “*Identification of CvkR binding site*” and the new Fig. 6).

To identify possible CvkR binding sites in the promoter regions of the genes beyond the CAST locus we searched for perfect matches to the improved CvkR motif 5'-AnnACATnATGTnnT-3' in the whole genome of *Anabaena* 7120. We found 8 hits, five on the chromosome, two on the delta plasmid and one on the beta plasmid. We can exclude a function for five of these hits as they are neither linked to any TSS in the vicinity nor conserved in related cyanobacteria. The three remaining motifs are within known promoters, including P_{cas12k} and P_{tnsB} . The third promoter drives the transcription of above-mentioned L-array. Unfortunately, the association of the CvkR motif with L-array promoters can be traced only to 25% of related cyanobacteria that have such an array and a CAST system. So, this association is rather weak and not mentioned in the manuscript, but it might be addressed in future work.

5. A large part of the paper discusses ATP binding interface, which authors believe is not a natural ligand. This makes this part of the paper inconclusive and vague. I suggest streamlining of this section.

Yes. Many thanks for your suggestion. We have revised this part as follows: “*In our solved structure, the crystal optimizing reagent ATP was bound near the putative effector binding site through π - π stacking, cation- π , and several hydrogen bonds (Fig. S6A). While the nonspecific hydrogen bonds at the N1 and N6 positions of the adenine moiety, together with the unbound triphosphate group in the putative effector binding site, suggest that ATP is not the natural effector of CvkR, other nucleoside derivatives might be possible candidates (Fig. S6B). Indeed, signaling molecules of*

the cyclic oligonucleotide family have been observed in certain types of CRISPR-Cas and other defense systems^{5,6}. However, subsequent EMSA experiments showed no effects on binding between CvkR and promoter DNA with ligands addition, indicating ATP and its analogues are indeed not the natural effector of CvkR (Fig. S7).” Please see also our response to point 7 of reviewer #2.

6. Related to 1. It is misleading to call non-homologous proteins of HTH and RHH family by the same name CvkR. For simplicity I suggest to keep the name only for MerR-like repressors, but do not assign it to RHH and other putative type V-K CAST regulators.

Yes. Many thanks for pointing this out. We have followed the suggestion and now use CvkR exclusively for the MerR-like regulators. We have made sure that all figures are labeled correspondingly.

In addition, we have renamed the regulators previously called “Omega- and CopG-like repressor” in our manuscript into Arc_1 and Arc_2, instead of formerly used more specific categories.

Minor corrections:

1. Line 70-71. I-F CAST encodes for five core proteins: Cas6, Cas7, Cas5, Cas8; the latter two are fused.

The passage has been corrected mentioning the Cas5-Cas8 fusion now.

2. Line 90. The logic behind this sentence is not clear: “Thus, the tight regulation of these systems can be expected.” A reference to previous work/reviews would help.

This paragraph has been re-worded.

3. Line 159. “The majority of cargo genes, located between cas12k and tniQ, are significantly more divergent”. What is the meaning of “divergent” here?

To be more precise, the sentence has been re-worded to: “The majority of cargo genes, located between cas12k and tniQ, are less conserved and vary in sequence and function.”

4. Figure 6B. Why the upstream -35 (blue) segment of promoter in the *Anabaena* 7120 is colored differently compared with identical segments in the alignment of other species? Same question applied for -10 region of CvkR. Explain better in the legend how these regions in other genomes were identified and colored.

The promoters in *Anabaena* 7120 were colored differently because these were determined experimentally, by RNA-sequencing, while the promoters in other species were identified with the program PromoterHunter^{7,8}. We wanted to visualize that difference, because the former information is more reliable than the latter. As we added the results of the experimental identification of the CvkR motif during

the revision of our manuscript (revised Fig. 5 & 6), this figure was replaced.

Reviewer #2 (Remarks to the Author):

The authors have identified a new MerR-family transcription regulator and proven/characterized the role of this protein as a repressor of class 2 type V-K CRISPR-associated transposase systems by extensive in vivo and in vitro experiments. In addition, a 1.5 Å crystal structure of the homodimer in complex with ATP was determined to provide structural insights, which is highly similar to other MerR family members except for some discrepancies in dimerization and effector binding domains. In general, although this study is not the first to discover/predict the existence of MerR-family regulators in CAST systems (see ref 14), it did provide solid data to identify and characterize the repressor role of this protein and the results will enrich the understanding of MerR-family transcription regulators. However, considering MerR-family members have been well and extensively studied and no novel mechanistic information on how MerR factors regulate transcription has been supplemented by this study, this reviewer has not been convinced that it is qualified for publication in Nature Communications and would suggest submitting it to a more specific journal.

The here provided revised version presents more information resulting from a series of additional experiments and analyses performed during the revision process. The data yield substantial novel insight into how this particular MerR-related factor regulates transcription and especially into the connection between CvkR regulators and CAST systems. For details, please see below.

Specific comments:

(1) It is one of the common phenotypes that a transcriptional regulator would regulate the transcription of nearby genes. Therefore, it is not surprising that CvkR would regulate the transcription of CRISPR system. The mechanism of this regulation is one of the most important questions. However, the authors have not shown a clear answer on how CvkR regulates this CRISPR system. For example, the signal triggering the conformational change of CvkR has not been demonstrated. The structures of CvkR with and without effector have not been compared. How does CvkR bind to promoter and influence the transcription carried out by RNA polymerase has not been investigated?

We quite agree with the reviewer that it is one of the most intriguing questions of our manuscript how CvkR regulates the CAST system.

From a bioinformatics perspective, we systematically analyzed the association of different regulators, including CvkR, with the existing CAST systems. The CAST systems consist of a CRISPR-Cas component that protect genome integrity and transposases, which

effectively have the opposite effect when transposition occurs. Thus, a tight regulatory control of the CAST systems can be expected. As suggested by reviewer #1, we were able to demonstrate a trend of horizontal gene transfer in *cas12k*, and the association of a certain type of regulator with a monophyletic group of Cas12k effector proteins by additional bioinformatic analysis.

From the perspective of genetic analysis, we provide solid data to support the repressor function of CvkR using multiple *in vivo* detection methods. During the revision of the manuscript, we further elucidated that CvkR directly controls the expression of *cas12k* and of transposase genes via the *tnsB* promoter, while it controls indirectly the abundance of the tracr-CRISPR RNA of the CAST system. We think these results clarify the biological function of CvkR.

Regarding our biochemical and cell-free analyses, we chose a more sensitive, chemiluminescent method to redo all the EMSA assays. In the TXTL assay, we identified a limitation that had restricted the evaluation of CvkR repressor activity in the first version of our manuscript. We found that the previously too high concentrations of CvkR led to constitutive binding activity and effectively repression of any promoter, including our now included new negative control promoter P_{psbAl} , and the not included control $P_{alr1654}$ here (see Figure 5F and Figure R1 below). We are now using much lower concentrations of 1 pM plasmid DNA throughout and have repeated all the previous TXTL experiments.

Figure R1. TXTL assays showing specific deGFP repression of the CAST related promoter P_{cas12k} with very low CvkR plasmid concentrations (right panel), while no repression was observed for the unrelated promoter $P_{alr1654}$ (left panel).

In addition, we performed another series of DNase I footprinting assays on the promoter of *tnsB*. We performed a larger number of EMSA assays using probes covering CvkR-controlled promoters, P_{cas12k} fragments and P_{cas12k} fragments with mutations. We finally identified the 15-nt CvkR motif AnnACATnTAGTnnT, which is conserved in potential CvkR regulons. Following your advice, we also made additional experiments regarding the possible effector(s) of

CvkR. As mentioned in our response to your point 7 in more detail, we tested several possible effectors, but none of them changed the DNA binding ability of CvkR in the used EMSA assays.

From the perspective of protein structure, we refined our CvkR structure model and made several additional superimposition analyses with other MerR-type regulators according to your suggestions. Based on these comparisons, we confirmed the distinct dimerization features. We also performed additional dynamic light scattering (DLS) analysis and found the hydrodynamic diameter of CvkR is ~5.1 nm (~51 Å), which is consistent with the size of the proposed CvkR homodimer. Following the advice of you and other reviewers, we generated CvkRmut (R19A-R20A-Q23A-K40A-R42A-N66A) and R42E mutants to obtain additional experimental evidence for verifying the role of these residues in the DNA binding by CvkR. In subsequent EMSAs, no shifted bands were observed for these two mutant proteins compared to CvkR wild type. Thus, both CvkRmut and R42E had lost the capability to specifically bind P_{cas12k} , directly supporting a role of the substituted residues in DNA binding (Fig. 8B). These results were also supported by the newly conducted TXTL experiments (Fig. 8D). All these data proposed a useful model for DNA interactions by CvkR.

In this study, we found P_{cas12k} is a typical MerR-regulated promoter with a prolonged spacer of 21 nt between the -35 and -10 elements. Thus, CvkR might modulate P_{cas12k} activity using the MerR-specific DNA-distortion mechanism. However, for P_{tnsB} we only find a -10/-35 spacer of 17 nt. As we demonstrated that CvkR binding in P_{tnsB} overlaps the TSS of P_{tnsB} , and partially the -10 element, CvkR should regulate the expression of *tnsB* in a different manner from *cas12k*. Beyond the direct regulation, CvkR impacts the abundance of the tracrRNA-CRISPR array transcripts likely indirectly, via governing the expression of Cas12k binding to these RNAs. Therefore, CvkR employs at least three distinct mechanisms to modulate the abundance of CAST components constituting substantial novelty of CvkR-mediated regulation.

(2) A structure of CvkR in complex with promoter DNA is needed to support the relevant presentation (lines 45-48, 411-428) in the manuscript. In the meantime, additional superimpositions with CueR repressor complex or other MerR-DNA repressor complex should be included, which will provide support for the description on the putative interaction between CvkR and promoter DNA. More specifically, lines 413-415: It's impossible to predict R42 is likely to participate in the binding of DNA based on Figure 9C. A figure of superimposing the structure of the previous MerR-DNA complex is necessary. The further description (lines 418-428) also needs relevant figure to support it.

Thanks. We attempted to solve the structure of the CvkR-promoter complex with great effort. Unfortunately, the resolution of the

diffraction data was only 6 Å, making it impossible to directly determine the structure of this complex. Considering that the wHTH domain, which mediates promoter recognition and binding in members of the MerR family, is highly conserved in structure, we resorted to structural alignments and biochemical experiments to investigate the regulatory mechanism of CvkR.

As CvkR shares several similarities with the MerR-type regulator HiNmIR (for instance, lacking a recognizable C-terminal sensor region and targeting an operon containing overlapping promoters), we first compared CvkR with the HiNmIR-promoter complex (5D8C)⁹. The well-matched wHTH domain showed that the positively charged enriched helix $\alpha 2$ and winged loop W1 separately insert into the major and minor grooves of DNA, while winged loop W2 is close to the DNA backbone. Subsequently, the key residues potentially involved in DNA binding of CvkR were further analyzed based on the structural positions of those identified in HiNmIR. The result showed that residues R20, Q23, and Y24 in $\alpha 2$, and residue N66 in W2 may be involved in the interactions between CvkR and its target promoter. Moreover, the positively charged residues R19, K40 and R42 in helix $\alpha 2$ and W1 are also possible to participate in DNA binding, especially R42, whose side chain simultaneously forms hydrogen bonds with the ATP's α -phosphate group (like in the DNA phosphate backbone) in our solved structure. The positively charged residues (frequently arginine) in the W1 region participate in target DNA binding in HiNmIR and some other MerR-family members. Thus, CvkRmut (R19A-R20A-Q23A-K40A-R42A-N66A) and R42E were constructed separately from each other. Subsequent EMSA and TXTL assays confirmed the role of these residues in the DNA binding of CvkR.

We have revised the corresponding sections in the manuscript and added the corresponding figures (Fig. 8B-D).

(3) About the structural model shown in the PDB validation report:
(i) Rfree value (23.1 %) is slightly low considering reporting a 1.5 Å X-ray crystal structure.

Yes, we agree with the reviewer. We re-processed our data to a 1.6 Å resolution and refined our structure model to a Rfree value of 21.8%. Meanwhile, we improved our crystal structure model to RSR Z-score of 16%.

(ii) RSRZ outliers (20.9 %) are too high. The real-space R-value (RSR) is a measure of the quality of fit between a part of an atomic model and the data in real space.

These suggest that the deposited pdb model needs to be improved.

We have improved the deposited pdb model and attached the updated structure validation report in our new submission in Table S5.

(4) Figures of SAD experimental map and final 2fofc map should be provided to show the quality.

Yes, we agree, the suggested data are provided in Figure R2 below.

Figure R2. The Experimental Electron Density Maps from SAD Phasing and final 2fofc map are displayed in gray. **A.** SAD experimental map. **B.** Final 2fofc map. The electron density of SeM/M57, 94, 98 is superimposed on the initial/refined model. The SeM/M57, 94, 98 are displayed as stick side chain model.

(5) In Figure 8, it's better to present superimpositions using homodimer structures to clearly show the distinct dimerization and effector binding domains in this protein. The current figure does not project these discrepancies.

Thanks for your productive suggestion. The new figure, Fig. 7E and F, and figure legends have been updated accordingly. To better present the comparison, we only selected two MerR-family regulators as representatives. CueR is a well-studied canonical MerR-type regulator, which assembles into a homodimer via two-helix antiparallel coiled-coil ($\alpha 5/\alpha 5'$) like most MerR-family members. GlnR is an unusual MerR-type TnrA/GlnR family regulator, which dimerizes relying on an N-terminal extra helix (e- $\alpha 1/e-\alpha 1'$). CvkR adopts a distant dimerization pattern, which is unique and novel among the MerR-family members. The corresponding section in the manuscript has been revised.

(6) ATP binding figure (Fig 9) is poorly presented. It's better to hide hydrogen when making figure and a clearer presentation is needed.

We appreciate this suggestion and the new figure (Fig. S6) has been updated.

(7) Authors obtained a structure of CvkR homodimer in complex with ATP, which binds at the putative effector binding region. Although the authors suggest cyclic oligonucleotide family molecules are the effector of CvkR, no candidates have been identified. This reviewer is curious what attempts have been tried. The possibility of cyclic oligonucleotide or even ATP as an effector could be easily tested. A control analysis with/without cyclic oligonucleotide/ATP in the in vitro DNA binding experiments (EMSA) should be considered and discussed. The reported structure may

not represent the repressor state. Please note that the available structures of CueR homodimer have three different states/conformations: CueR dimer with effector Ag/Cu(I) (pdb: 1q07), CueR-DNA repressor complex (pdb: 4wls), and CueR-DNA activator complex (pdb: 4wlv).

To identify the effector molecule, we tested CvkR's ability to bind P_{cas12k} in the presence of multiple molecules with similar chemical structures to ATP. We performed EMSA assays with CvkR (0.5 μ M) and molecule candidates (adenine, adenosine, AMP, ADP, ATP, cAMP, guanine, guanosine, GMP, GDP, GTP, cGMP). The DNA binding ability of CvkR was not changed by any of these molecules (novel Fig. S7A).

To further test if higher concentration of these candidates affects the protein-DNA binding, we also performed EMSA assays with concentration gradients of ATP, GTP, CTP, cAMP, and cGMP (from 0.25 mM to 5 mM). We found no altered protein-DNA binding under these conditions as well (novel Fig. S7B).

The relative orientation of two DNA-interacting helices $\alpha 2'$ and $\alpha 2$ was found to be able to reflect the activator/repressor state of both CueR and HiNmIR in their structural analysis^{9,10}. Based on these observations, we performed further structural comparison among CvkR and CueRs in different states (1Q07, 4WLS, 4WLV) following your advice. The results indicate this relative orientation of $\alpha 2'$ and $\alpha 2$ in CvkR is more similar to that of CueR's activator state rather than its or HiNmIR's repressor states. These findings suggest that our solved CvkR structure is more resembling an activator state or an approximative activator state, even though ATP is not the natural effector (Fig. R3). Nevertheless, this conclusion might be taken with caution because the CvkR dimerization unit radically differs from other MerR-family dimerization units.

Figure R3. Comparison of the relative positions of the DNA-interacting helices $\alpha 2'$ and $\alpha 2$ for MerR-type regulators. wHTH domain from one subunit is used for superimposition, and the relative position of helix $\alpha 2'$ in the other subunit following superimposition is boxed. Superimposed regulators: CvkR (this work; coloring in blue), repressor complex for CueR (PDB code 4WLS, coloring in red), activator complex for CueR (PDB code 4WLV, coloring in green), CueR-effector complex (PDB code 1Q07,

coloring in orange), and HiNmIR-promoter DNA complex in repressor state (PDB code 5D8C, coloring in cyan). The wHTH domain from one subunit and the helix $\alpha 2'$ from the other subunit are shown as ribbons, and additional regions are omitted for clarity.

(8) Line358-361: Unit of contact area should use \AA^2 . Contact area analysis in this study was calculated using the interface between one CvkR molecule in an asymmetric unit and its symmetric mate. Are there other possible dimerization interfaces found in the crystal packing? If so, what are the other interface areas? Smaller than 970.1 \AA or not? What are the contact areas of dimerization in other MerR factors? In addition, this contact area can only suggest stronger/weaker interaction between two CvkR molecules. The dimerization state of this protein in solution should be determined by other methods, such as SEC and/or light scattering analysis. The relevant sentences need to be modified.

Thanks. We have revised the unit of contact area to \AA^2 . Our earlier SEC assay has shown that CvkR is a homodimer in solution. Under this premise, we analyzed its assembly in our solved CvkR structure. Four neighboring CvkR molecules as the potential dimeric partners were found in the crystal packing (Figure R4A, B, C, and D), and the dimeric architecture only in Figure R4A is similar to other reported MerR-family members. The interface areas of these four homodimer forms are 997.6 \AA^2 , 843.3 \AA^2 , 372.2 \AA^2 , and 276.6 \AA^2 , respectively, analyzed through webtool PDBePISA¹¹. In addition, PISA analysis shows that the assembled dimer form of CvkR presented in our manuscript (A form) is the only reasonable solution.

Next, besides the SEC analysis, we performed the suggested dynamic light scattering (DLS) analysis during the revision of our manuscript. The results showed the average diameter of CvkR in solution is ~5.1 nm (~51 \AA), which is consistent with the size of homodimer in A form (~54.3 \AA). The measured sizes in B (~72.8 \AA), C (~73.5 \AA) and D (~92 \AA) are all much larger than 51 \AA . To sum up, CvkR forms a homodimer in a style as displayed in Figure R4A. Three antiparallel β -strands ($\beta 2$ - $\beta 1$ - $\beta 3$) and one short α -helix ($\alpha 6$), together with the equivalent β -strands and helix of the other subunit, comprise the bulk of the dimerization interface. The contacts are predominantly hydrophobic and bury 997.6 \AA^2 surface area (PDBePISA), which is much smaller than that observed in canonical MerR-type proteins, for instance, 1896.8 \AA^2 in CueR. The corresponding section in the manuscript has been revised.

Figure R4. Analyzing the potential dimeric partner of CvkR in the crystal packing.

(9) A superdex 200 10/300 column used in the study is not a good one to determine the dimeric or monomeric state of CvkR (~17 kDa). A superdex 75 10/300 column and light scattering analysis should be utilized.

All the SEC analyses were re-performed using the superdex 75 10/300 column according to your suggestion.

We verified that CvkR truly forms a homodimer in solution. In addition, the homodimer state of all the newly constructed CvkR mutant proteins was analyzed using the same column as well. As mentioned above, dynamic light scattering analysis (DLS) was also performed following your advice.

(10) Line429 6 aa were selected to be mutated to generate the CvkRmut to study the relevance of DNA binding and transcriptional regulation. It will be good to first analyze its oligomeric state in solution using SEC and then conduct a DNA binding ability assay (EMSA) before the final TXTL functional assays.

We quite agree with you. We have performed the SEC and EMSA analyses according to your suggestion.

(11) Mutations have been constructed in the putative DNA binding domain. Since the dimerization domain is another distinct region of its structure, this reviewer suggests a relevant mutagenesis study on this region to strengthen the relevant statements.

As mentioned in the response to point 8 of this reviewer, after a careful structure analysis, we found it is mainly the hydrophobic interactions that contribute to the dimer formation of CvkR. Thus, it is difficult to disrupt this interface by mutating only a few residues. In addition to the structural analysis, both DLS experiments and PISA analysis support our proposed homodimer interface as well.

(12) The location of CvkR binding region in the target promoter is important for explaining the regulatory mechanism. Authors may present the CvkR binding region and promoter -10 and -35 elements in a main figure.

Many thanks for this suggestion. We have performed additional experiments and identified a 15 nt CvkR binding motif. These elements are now included in the revised Fig. 6 of the main text.

(13) The classical MerR family regulators (such as CueR, BmrR) have been demonstrated as activator, but some proteins have been shown as repressor (McdR, HonC). Authors may discuss the potential mechanism of CvkR by comparing with these regulators.

Many thanks for this suggestion. We discuss now the potential mechanism of CvkR by comparing with the mentioned regulators in the second half of our discussion.

(14) Authors demonstrated that CvkR could be translated from leaderless mRNA, but did not show the meaning of this leaderless mRNA translation. It might be interesting to test if the functions of CvkR-L and CvkR-S are different.

Transcriptomic data and western blot analyses exclude the possibility of CvkR-L existing *in vivo* in *Anabaena* sp. PCC 7120 wild type under all tested culture conditions.

Leaderless gene expression is an evolutionarily conserved function of translation initiation. The special class of leaderless mRNAs has been found in all three domains of life. However, compared to eukaryotes and archaea, the frequency of leaderless mRNAs in bacteria, especially that in gram-negative bacteria, is rather low¹²⁻¹⁴. In the cyanobacterium *Synechocystis* sp. PCC 6803, previously 51 instances of leaderless mRNAs were reported¹⁵. Even in the extensively studied model organism *E. coli*, deep sequencing only found less than 30 leaderless mRNAs in its whole transcriptome under standard growth conditions¹⁶. As such, the biological function of leaderless gene expression largely remains to be established. Tn1721 *tetR* mRNA and λ *cI* mRNA are the best studied leaderless mRNAs derived from transposons and bacteriophages in *E. coli*. These leaderless mRNAs were proposed as mediators of horizontal gene transfer controlled at the translational level¹³ and this might apply to CvkR too, especially as it was identified in this study as regulator of the CAST system, which has ties to transposons and phages as well. However, the real relevance of leaderless mRNA translation in this cyanobacterial CAST systems needs to be further studied.

Reviewer #3 (Remarks to the Author):

In the manuscript, the authors report a transcription regulation mechanism for repressing the basal activity of CRISPR-associated Transposons (CASTs). The

transcription repressor, CvkR of CAST in *Anabaena* sp. PCC 7120 was chosen to study the detailed repression mechanism. The study shows that CvkR represses the expression of cas12k as well as other genes encoding essential components of CAST. An long DNA-recognition motif of CvkR was identified in the core promoter region of cas12k and a high-resolution crystal structure of CvkR is reported. Overall, the manuscript explains how bacterial cells restrict the basal activity of CAST by showing that CAST is transcriptionally repressed by a new MerR-type transcription factor. However, more evidence should be collected to demonstrate the specific binding of CvkR to its predicted binding motif and more structural data, ideally a crystal structure of CvkR-DNA binary complex, should be provided to explain how CvkR recognizes the long DNA binding motif. The detailed comments are listed below.

1. Please add figure citations in the second and third paragraph on Page 6 and the first paragraph on page 7.

Done as suggested.

2. Figure 2. The authors reported that at least one regulator near cas12k gene was identified in 94 CASTs. Please clarify how many CASTs were surveyed.

We had 118 analyzed, number was added to Figure legend.

3. Fig 3B does not add convincing argument for the shorter ORF of CvkR-S.

Many thanks for pointing this out. The figure is only used to display the distance between the regulator genes and cas12k genes. We now describe it in the text as follows: "A generally shorter distance between these two cognate genes than with other associated regulators was also observed for other systems."

4. The distance between CvkR and Cas12k genes are less than 100 nt (Fig. 3B), does the binding of CvkR on its predicted cis element affects the expression of cvkR itself?

The promoter of *cvkR* (P_{cvkR}) was not able to drive deGFP transcription in the TXTL system (Fig. 5E). Therefore, whether CvkR can regulate its own transcription could not be tested in this assay. If the overlapping *cas12k* and *cvkR* promoters were separated into two fragments, we did not observe CvkR binding to the P45 fragment (Fig. 5C) containing the core region of P_{cvkR} including predicted -10 and -35 elements. In contrast, strong binding of CvkR was observed to the P43 and P_{wt} fragments containing the core region of P_{cas12k} in the EMSA assay (Fig. 5C & 6D). These results point away from a possible autoregulation of CvkR.

5. Fig. S1C. Please add the explanation for the yellow and red bars in the figure legends.

Thanks for the suggestion. The red bar indicates the extra 54-bp fragment at the 5'-end of *cvkR-L* compared to *cvkR-S*. The yellow bar

indicates the 31 nt long 5'-UTR of *petE*. We have added the description in the revised version.

6. Fig. 3c. Please explain the WT. I suppose it means the *cvkR* depletion strain but not the genuine 'wild-type' strain.

Here it indeed indicates the *Anabaena* sp. PCC 7120 wild type strain. There is nearly no signal of *cvkR* mRNA in *Anabaena* sp. PCC 7120 wild type under normal culture conditions. This was also confirmed by our northern hybridization and microarray assays. In addition, the results were further verified by the western blot on protein synthesis level.

7. Fig. 3D shows CvkR-L can be translated under control of an artificial promoter, thereby did not provide support for the only existence of CvkR-S in bacterial cells. The author states that 'the start codon of *alr3614S* coincides with the previously mapped TTS of its mRNA' (line 204). This is a better argument and deserves a supplemental figure showing the mapped TTS, start codon, the predicted -35/-10 elements of the *cvkR* promoter.

Thanks a lot for the suggestions. We have inserted the Fig. 3B to show the mentioned elements.

8. Fig 4B-E. Please label the positions of the target bands. In Fig. 4B, I did not see much difference between the wt and *cvkR* depletion strains. Does that mean the expression of *cvkR* gene is repressed in wt bacterial cells? In Fig. 4D, What is the identity of the bands of ~ 500 nt that showed strong signals in wt bacterial cells but were absent in *cvkR* depletion and *cvkR* complementation strains. Fig S2 suggests that the *tracrRNA*-CRISPR array has its own promoter. How would it explain *cvkR* depletion also increases the transcription level of *tracrRNA* and CRISPR? Is there a CvkR degenerate binding motif of CvkR on that promoter? Did the authors test whether CvkR interacts with the promoter in a sequence-specific manner?

Done as suggested. Positions of the target bands are labeled by triangles.

Fig. 4B: The results of northern hybridization, in combination with those of microarray assay, qRT-PCR, as well as the western blot on the protein level, suggested that the amount of *cvkR* mRNA and CvkR protein is close to or below the detection limit in *Anabaena* sp. PCC 7120 wild type cells under standard growth conditions. However, *cvkR* deletion indeed led to obvious signal changes for *cas12k*, *tnsB* and *tracrRNA*. These suggested that CvkR likely is expressed at a low level but sufficient to perform the repressor function in *Anabaena* sp. PCC 7120 wild type.

Fig. 4D: There were no bands of ~ 500 nt that showed strong signals in wt bacterial cells but were absent in *cvkR* depletion and *cvkR* complementation strains in Fig. 4D. We assume the reviewer is asking regarding Fig. 4C. These bands of ~ 500 nt (>400 nt) come from the joint precursor of the *tracrRNA*-CRISPR. When hybridized with the

CRISPR array probe, we observed the mentioned strong signal of the joint precursor in WT but less so in $\Delta cvkR$. Deletion of *cvkR* should lead to the overexpression of the joint precursor, however, this joint precursor is subsequently processed into the major accumulating fragments of ~150 (CRISPR array) and ~200 nt (tracrRNA), respectively. Thus, the CRISPR array showed much more abundant signals in $\Delta cvkR$ compared to WT and $\Delta cvkRCom$ (Fig. 4C), while tracrRNA was also well detectable in WT and $\Delta cvkRCom$ (Fig. 4D). We have revised the relevant descriptions in the new version to make the explanation more clearly.

Regarding previous Fig S2: Yes, the tracrRNA-CRISPR array is co-transcribed. Deletion of *cvkR* led to the upregulation of both tracrRNA and CRISPR array-derived transcripts. We didn't find any possible CvkR binding motif in the corresponding P_{tracr} promoter. Following your suggestion, we performed EMSA and TXTL assays and found no binding of CvkR to the promoter fragment. Thus, we think CvkR influences the expression of the tracrRNA-CRISPR array indirectly, through other factors. The most likely candidate for such a factor would be Cas12k. In this scenario, *cvkR* deletion increases the transcript level of tracrRNA and CRISPR via the enhanced accumulation of Cas12k that stabilizes the transcript as an RNA binding protein. We have added these considerations and the mentioned new results in the revised manuscript.

9. Fig. 5C show TnsB is also upregulated upon depletion of *cvkR*. Is there a degenerate CvkR binding motif on TnsB promoter? Did the authors test whether CvkR interact with the promoter in a sequence-specific manner?

Yes, we performed additional DNase I footprinting assay with *tnsB* promoter during the revision of our manuscript, and found a CvkR motif within the *tnsB* promoter, P_{tnsB} . Furthermore, we also performed EMSA and TXTL assays and confirmed the direct sequence-specific binding of CvkR to P_{tnsB} . Thus, the *tnsB* promoter is indeed under CvkR control. Please see the revised Fig. 5 and 6 in our manuscript.

10. Please indicate the TSS in Fig. S2A and S2B

Only the TSSs of *tnsB* and tracrRNA of *Anabaena* sp. PCC 7120 were identified. We have revised this figure and labeled the TSS in the revised version (now in Fig. 6B). The previous Figures S2A and B with the multiple sequence alignments are now shown as Fig. S1B and S4.

11. Please label the target *cvkR* band in Fig. 5A.

Done as suggested (now Fig. S3).

12. In Fig. 7A, the result is ambiguous. The presence of 5 nM CvkR shifted majority of DNA in all tested promoters, while the presence of lower concentration of 1.5 or 1 nM CvkR shifted DNA in a similar extent in all tested promoters. Moreover, it is better to

include point mutations (or combination of point mutations) of the CvkR motif in the EMSA and deGFP reporter assays to validate the binding motif.

Many thanks for the suggestions. In the previous version of our manuscript, the EMSA assays were performed using the ethidium bromide staining-based method. To improve sensitivity and get convincing evidence, the whole EMSA assays were re-performed using the biotin-labeled chemiluminescent method. Meanwhile, we performed additional DNase I footprinting assay and EMSA assays with probes containing point mutations and combination of point mutations, and finally identified a 15 nt, more precise motif. We have added these new results in the revised manuscript.

The concentrations of CvkR used in the TXTL assay in the previous manuscript version were too high. Therefore, this section was revised entirely. Please see also the more detailed reply to point 13 just below.

13. In Fig. 7F, 5 nM CvkR is used to repress *PtracrRNA* instead of 1nM CvkR as used in Fig. 7D and 7E. Does higher CvkR protein level cause non-specific DNA interaction and thus non-specific transcription repression? A control should be included to prove that neither 1nM nor 5nM CvkR repress transcription from a non-relevant promoter.

We did test also 1 nM CvkR as was used in Fig. 7D and 7E but decided in the original version to show only one dataset. During the revision we have repeated this analysis including several more concentrations and now included two controls of non-relevant promoters as proposed.

We tested several CvkR plasmid concentrations and figured out that the initially used concentrations were too high (Figure R5 below). We identified 1 pM to be a good CvkR plasmid concentration to see specific repression (Figure R1). We repeated all experiments using 1 pM CvkR plasmid. With this concentration, the *cas12k* promoter was repressed completely but the *tracrRNA* promoter was not (Fig. 5E). Additionally, we tested the *tnsB* promoter (Fig. 5E) that was completely repressed by CvkR. We also tested a poorly expressed negative control ($P_{alr1654}$, see Figure R1, not included in the revised manuscript) and a highly expressed negative control (P_{psbA1} , Figure R6), that we included in the manuscript in the revised Fig. 5F.

Figure R5. Using higher CvkR plasmid concentrations deGFP expression driven by P_{tracr} is repressed by constitutive binding.

Figure R6. Reporter gene assays. The full-length versions of P_{cas12k} and P_{psbAI} were tested in the TXTL system for their capacity to drive deGFP expression and mediate repression upon parallel expression of CvkR. The P_{psbA1} promoter is not controlled by CvkR. One pM CvkR plasmid was expressed together with the corresponding p70a plasmids (5 nM) with the respective promoter variants upstream of the deGFP gene. Error bars show standard deviations calculated from 2 technical replicates.

14. Line 359, '970.1 Å' should be '970.1 Å²'

Many thanks. Revised as suggested.

15. Fig. S3, please add the calculated molecular weight in the figure legend.

Yes, we have re-performed the SEC analysis using the superdex 75 10/300 column (according to the suggestion of Reviewer #2) and generated the Fig. 5A with the calculated molecular weight added in the figure legend according to this comment by Reviewer #3. This was also done for the CvkR mutant proteins in our revised manuscript (Fig. S8).

16. The two half palindromic sites of the CvkR motif (5'-AAAACACA-N21-TGTGTTTT-3') is separated by a 21-bp spacer, which is much larger than the spacer length of typical MerR-family TF motifs, and much larger than the spacer length of binding motifs of other bacterial TFs. To recognize the two half palindromic sites spanned by 29 bp (21+4+4) simultaneously by the CvkR homodimer, The DBDs of the two CvkR protomers should be separated by ~100 Å, much larger than the current distance of the two DBDs in the crystal structure (28 Å; Fig. 8B). Therefore, the CvkR dimer must undergo drastic conformational change upon DNA interaction. It is unknown whether and how the CvkR dimer is fully stretched to bind its long motif. A crystal structure of DNA-bound CvkR would explain how CvkR interacts with the long motif.

Thanks for pointing out this problem. We quite agree with you. We have realized that this motif was too long to be a typical binding motif of a MerR-family transcription factor. As also mentioned by Reviewer #3 (point 6), the oligomeric adenine tracts at both ends of this motif on the DNA double strand might not be good DNase I targets. To further solve the problem, we first carried out EMSA assays with diverse probes targeting the key nucleotides involved in the potential palindrome structure (Fig. R7). Compared to probe P_{cas12kcenter} (the CvkR-protected 43-nt of P_{cas12k} in the DNase I footprinting assays), we found a 5-nt insertion (probe -35M) or deletion (probe -35Mm5) in the spacer has no obvious effect on CvkR binding. Furthermore, CvkR was detected to bind specifically to the probe IR1Am (mutation in the entire left half side of the long motif covering the A tracts) but with no affinity of CvkR to the probes IR2Tm (mutation in the entire right half side of the long motif covering the A tracts) and IR12ATm2 (mutation in both half sides of the long motif). All these results indicate that the key CvkR binding sites are located in the right part rather than the left part of P_{cas12kcenter}. We also designed probes in which only the oligomeric adenine and thymidine tracts were mutated. The strikingly different retardation signals between probe IR12ATm and IR12ATm1 indicates that the oligomeric thymidine tracts are involved in the formation of the core CvkR motif, especially the second "T" (Fig. R7).

Figure R7. EMSA assay of previously proposed long CvkR motif. A. Wild-type and mutated promoter fragments used for EMSA assays. TSS, -10, and -35 elements in $P_{cas12kcenter}$ are highlight in red, green, and blue respectively. The mutant sites are highlighted in pink letters. **B.** Interaction between CvkR and $P_{cas12kcenter}$ variants analyzed by EMSA.

We then performed an additional DNase I footprinting assay with the promoter of *tnsB* and found a 41-nt fragment that was protected by CvkR ($P_{tnsBcenter}$). Intriguingly, a near-perfect matched 16-nt fragment (5'-ATAACATTATGTRTTT-3') was observed between $P_{cas12kcenter}$ and $P_{tnsBcenter}$. The only discrepancy between them is the G/A substitution at the 13th position (Fig. 7A). Thus, this fragment would be in the core of a potential candidate of the CvkR binding motif. By further screening inside this fragment, we found a 15-nt motif with perfect invert repeats (5'-AnnACATnATGTnnT-3'). We eventually confirmed this motif by performing a series of EMSA assays using probes with mutations on the palindrome and the spacer, both sequence and length (Fig. 6C and D). In addition, this CvkR motif was supported by the bioinformatic analysis (Fig. 6B). We also made structural superimpositions of CvkR to other MerR-type regulators with resolved protein-DNA structures. Based on these additional analyses, we were able to further demonstrate the mechanism of CvkR-promoter binding.

17. Line 382 and Figure 8c, please add citations for the CueR and SoxR structures used for comparison.

Done as suggested.

18. Line 393, 'Notably, the hydrophobic residue W133, originally embedded in the hydrophobic interior, is exposed to the solution side due to the binding of ATP'. The authors compared the W133 conformation in the presence or absence of ATP binding.

Which structure was used to represent the W133 conformation in the absence of ligand?

Thanks for pointing out the inaccuracies in our previous descriptions. We have revised them. In addition, as ATP has been verified not to be the natural effector of CvkR, to avoid confusion, we have streamlined the section on ATP binding interface analyses as suggested in point 5 of reviewer #1.

19. Lines 416-417, It is better to delete the sentence. Residues making interactions with ATP does not necessarily indicate its capacity of interaction with DNA.

Done as suggested.

20. Line 418-428, please prepare a supplementary figure to show the structure superimposition.

Yes, a new Fig. 8A has been provided to assist in describing the DNA recognition of CvkR.

21. Fig 10C. It is surprising that a combination of six point mutations didn't abolish the sequence-specific DNA recognition by CvkR (5nM). Is it because other potential key DNA-contacting residues are not included in the point mutations?

During the revision we carefully addressed this problem and found that the previously too high concentrations of CvkR or CvkRmut (with six-point mutation) led to nonspecific binding activity and effectively repress any promoter. This should be the main reason for the transcriptional repression in the TXTL assay in the original version. We also have tested the CvkRmut again at our now established lowered plasmid amount of 1 pM in the TXTL assay. The ability of this protein to repress is impaired to a large extent. Meanwhile, we also have performed additional EMSA assays and found that the CvkRmut protein with six-point mutation entirely lost the promoter binding ability *in vitro*. Please see also the response to point 20 directly above.

22. Line 484, recent structural works of transcription activation complexes the MerR-TFs should be cited.

Yes, done as suggested.

Reviewer #4 (Remarks to the Author):

This manuscript by Ziemann and colleagues describes a series of phylogenetic, genomic, cellular, structural and molecular studies on CvkR from the cyanobacterium, *Anabaena* sp. PCC 7120. CvkR is the authors suggestion and is very reasonable. Their data reveal that CvkR is a repressor of the class 2 type V-K CRISPR-associated transposase system. Moreover, their structural work reveals a new subfamily of the

MerR superfamily. This study is very interesting and for the greater part the work is well done. However, there are several issues that the authors should address.

Many thanks for these encouraging and positive comments! They are highly appreciated.

1. The authors refer to the “novel” C-terminal domain, with a standard alpha helix ($\alpha 7$) and lacking the cysteine that is conserved in the other CvkR family members, which the authors identify in their phylogenetic studies. I agree that the C-terminal domain and the dimerisation mechanism is new, but the importance of helix $\alpha 7$ is unclear with respect to effector binding. The authors should change the serine found in their CvkR protein to a cysteine to assess its importance. From reading the manuscript, the authors imply this change is important, but provide no evidence to support this implication. Also, the authors should consider removing $\alpha 7$ and test the functionality of the resulting CvkR $\Delta 7$.

Following this advice, we removed $\alpha 7$ and tested the functionality of the resulting SUMO-CvkR $\Delta 7$ in the TXTL assay. The transcriptional repression function was clearly impaired (Fig. R8). CvkR $\Delta 7$ protein precipitated immediately when the SUMO tag was removed. Thus, we were not able to test the DNA binding ability of this mutant in parallel with other untagged CvkR mutants.

Fig. R8 TXTL assays to test the regulatory capacity of CvkR $\Delta 7$. **A.** Western blot analysis of CvkR $\Delta 7$ via their N-terminal 6xHis tag. **B.** TXTL assay with CvkR $\Delta 7$.

Different from the N-terminal DNA binding domain of MerR-type regulators, the C-terminal effector binding domains are not conserved, vary both in sequence and length. The metal- and redox-responsive MerR-type regulators usually harbor a very short C-terminus. A typical feature of this MerR subfamily is that nearly all members contain conserved cysteine residues within the C terminal domain, such as CueR and SoxR of *E. coli* and CadR of *Pseudomonas*

***putida*. The number and position of these cysteine residues play vital roles in defining the coordination geometry of the effectors. In contrast, the C-terminal domain of the drug-resistance MerR regulators is much larger than that of the metal- and redox-responsive MerR subfamily. Though the C-terminal length of CvkR is comparable to the latter MerR members, both our sequence alignment and crystal structure analyses revealed specific features within the effector-binding domains of CvkR. We found one of the conserved cysteines (at position 134, Fig. S5 and S9) is replaced by serine in CvkR, whereas S134C mutant presents similar behavior to the CvkR wild type in SEC, EMSA, and TXTL experiments (Fig. S9). Moreover, the cysteines of CvkR are structurally dispersed distributed, and all point far away from the putative effector binding site based on the structural superimposition (Fig. S5). These findings imply the specific effector binding of CvkR is likely conducted differently from the conventional cysteine-dependent manner as found in the metal- and redox-responsive MerR regulators.**

2. A larger issue that the authors must address is the inclusion and discussion of other MerR families. They completely ignore TnrA and GlnR, which form a different branch of the MerR family. Indeed, these MerR family members have very distinct C-termini and N-termini that are involved in dimerization. Further, these proteins bind protein to effect their transcription regulation. They also do not bind the canonical MerR DNA binding site, i.e., one in which the -10 and -35 boxes are separated by at least 18 base pairs. The authors also fail to include any discussion of BldC, which is a critical regulator involved in development and oligomerises and binds DNA in a fashion different to MerR, BmrR, CueR etc. and likely CvkR. A more in-depth discussion is these other MerR proteins is necessary.

Yes. Many thanks for your suggestion. Following your advice, we added the structural comparison between CvkR and GlnR (Fig. 7F in the revised manuscript), which further highlights the unique feature of CvkR dimerization. Meanwhile, we also discuss the potential mechanism of CvkR by comparing with BldC as well as several other MerR-type regulators in the second half of our discussion.

3. The authors present a DNA binding site but do not measure the affinity of CvkR for this site. From the presented EMSA experiments, the affinity would not appear to be that high. They should determine the affinity and follow this up with mutation of the palindrome and the spacer, both sequence and length.

Many thanks for this suggestion. As mentioned in our response to point 4 of reviewer #1, point 1 of reviewer #2, as well as point 16 of reviewer #3, we have performed additional DNase I footprinting assay and bioinformatic analysis to specify the CvkR binding motif further. Meanwhile, we have utilized the more sensitive biotin-labelled chemiluminescent EMSA method and carried out all the experiments following your advice. Based on these efforts, we eventually identified

the CvkR motif as 5'-AnnACATnATGTnnT-3'. We have added these new results in the revised manuscript.

4. The crystal structure was obtained to high resolution but required the presence of ATP in the crystallisation drop. On page 16 the authors describe the ATP binding site and the interactions. This is not well done. As the specificity for the adenine base is not described fully. From Figure 9, there is no way to tell how the N6 (hydrogen bond donor) and N1 (hydrogen bond acceptor) of the ring are “read” by the protein. It looks like the peptide backbone is involved. Further, the pi-pi and cation-pi interactions contribute to affinity but not to specificity. The authors should mutate at least residue R136 to understand its importance in ATP binding (see below). Moreover, the authors write, “Notably, the hydrophobic residue W133, originally embedded in the hydrophobic interior, is exposed to the solution side due to the binding of ATP...”. First, W133 is aromatic, not hydrophobic, and as such this residue type can be found on the surface of a protein. More, how do the authors know that W133 is “originally embedded in the hydrophobic interior”?

We appreciate this valuable suggestion. The hydrogen bonds at N1 and N6 are indeed formed with the peptide backbone, implying the binding of adenine moiety is not specific. As for the inaccurate description on W133, we have corrected it. Since ATP is not the real effector, which is confirmed by EMSA assays in this revision. The construction of an R136 mutant seems to be meaningless. In the meantime, we have streamlined the ATP binding interface analysis to avoid confusion following the advice of point 5 of reviewer #1. The corresponding section has been updated in the revised manuscript.

5. The authors should measure the affinity of the ATP for CvkR as well as the affinities of ADP and AMP. They should also measure the affinities of CTP, which could present N4 (hydrogen bond donor) and N3 (hydrogen bond acceptor) to the protein, and GTP, which should not be compatible because it would present O6 (hydrogen bond acceptor) and N1 (hydrogen bond donor).

We have evaluated the potential of these molecules as CvkR effectors by EMSA experiments and confirmed none of them is the natural effector of CvkR (see new Fig. S7). In this scenario, we decided not to measure the affinities of these molecules for CvkR. The related results have been supplemented in the manuscript.

6. The authors identify a palindrome-containing DNA binding site via DNase I protection. There are four adenines on one end and four on the other end. Although these A tracts are possibly part of the CvkR cognate site, DNase I does not do a very good job in cutting these types of sequences. The suggestion here would be just to couch the wording that the A tracts are not always cut by DNase I.

We appreciate this valuable suggestion. As mentioned in our response to point 3 of reviewer #4, we carried out an additional DNase I footprinting assay with promoter of *tnsB* gene to further identify the CvkR motif. We indeed found these A tracts might not be good targets of DNase I in our experiments.

We have introduced the improved motif we confirmed by both DNase I footprinting and EMSA assays in the revised version.

7. Page 19: Again, in the Discussion there is nothing about GlnR/TnrA or BldC. Also, the authors should include the name(s) of “MerR-like proteins without effector interactions...”. MtaN is probably also in this category.

Thanks, we improved both the results and discussion sections by adding a comparative analysis of CvkR with other MerR-family members. These include CueR, GlnR, BldC and McdR^{17–20}. We discuss the regulation mechanism of CvkR as well as the unique dimerization pattern and distant C-terminal packing of CvkR in the result section.

As for MtaN, it’s a 109 residual truncation mutant that lacks the C-terminal effector-binding sensor domain of the multidrug transporter gene activator Mta²¹. The size and function of the full-length Mta seem to be distant from that of CvkR. Besides, the dimerization pattern of MtaN is consistent with CueR, the MerR regulator we compared to CvkR extensively in this study. Thus, we selected HiNmIR, which lacks a recognizable sensor region like CvkR, for the sensor and regulation mechanism analyses in this study.

The corresponding content has been added to the revised manuscript.

8. Page 21: “the efficient binding of an adenine points to a metabolite that may be related to the cyclic nucleotide family of signalling molecules...”. What does efficient mean here? You have added 10 mM ATP as a crystallisation reagent. Without a K_d, this is not an appropriate statement.

The respective half sentence has been deleted and the rest of the sentence re-worded avoiding this overstatement now.

9. Page 27, line 674: “...0.3 mg CvkR...”. Please provide the concentration, not the amount.

Thanks for the suggestion. We have changed it into “five hundred microliter proteins (~0.6 mg mL⁻¹)”

10. Page 27, line 678: The names and masses of each molecular weight marker should be given here and included in the Supplementary Figure S3.

Thank for the suggestion. We have added the names and masses of each molecular weight marker in the Materials and methods section. We also marked the proper markers in the new Fig. 6A.

11. Figure 2 suggestion: It would be helpful if the authors could label the top of the columns with something like RT, DBD, O. This would make it easier for the reader to relate the data to the Repressor Type, DNA-binding Domain and Organism.

Done as suggested.

12. Figure 8C: The authors do not really provide much of a discussion concerning the overlays that are presented. Furthermore, they did not include other MerR family members in the overlays including TnrA/GlnR, BldC or BmrR.

Many thanks for pointing out this problem. To better interpret the unique feature of CvkR dimerization, we removed previous Fig. 8C and generated Fig. 7E and 7F in the current version using two typical and well-studied MerR-type regulators for comparison. In an overall view, the mode of CvkR dimerization is significantly different from CueR, which assemble into homodimers via two-helix antiparallel coiled-coil ($\alpha 5/\alpha 5'$) (Fig. 7E), and is also distinct from unusual MerR-type TnrA/GlnR family regulators, which form homodimer relying on an N-terminal extra helix ($e-\alpha 1/e-\alpha 1'$)²² (Fig. 7F). Therefore, the pattern of CvkR dimerization is unique and novel among the MerR-family members.

Meanwhile, we discussed the regulation mechanism of CvkR as well as the unique dimerization pattern and distant C-terminal packing of CvkR with several other MerR-type regulators as mentioned in our response to your point 7.

13. Methods: In several instances the authors refer to previously published work for a description of the methods that they employ in this work. They should include more information as this is a bit of a disservice and inconvenience to the reader.

Substantially more details are provided now. We extended the sections on microarray analysis, on Northern and on Western blot analyses.

Minor:

14. Page 14, line 347: Suggested word change: "The results shown in Figs. 4 to 7..." might be better stated as "Our data has established..."

Done as suggested.

15. Page 16, line 389: "binds exactly to the putative effector-binding domain...". What does exactly mean here? This is not likely the best choice of adverb.

Sentence was modified.

16. Page 19, line 467: The authors should refer to the original MerR papers by Walsh and Summers, not just a reference to two general reviews.

Reference was added and the reviews shifted to a more appropriate place.

17. Page 19, line 478: "...so MerR represses the binding of the σ factor." This is not quite correct. The typical repressive MerR proteins block σ factor binding by altering the DNA conformation around the -10 and -35 promoter elements. The work repress implies something else.

Many thanks for pointing this out! Sentence was corrected accordingly.

18. Page 29, line 717: "...15%-20% PEG300." Do the authors mean PEG3000?

Thanks. We carefully checked the experimental conditions and confirmed that the crystallization reagent used was indeed PEG300, not PEG3000.

19. Figure 4: The authors should point out the differences in the bp ladders between B-D and E.

We have used 3 different size markers with slightly different compositions of bands. Low range RNA ladders are used for high resolution polyacrylamide gels separating small RNAs up to 500 nt, here in panels B to D. High Range RNA ladders are used for agarose gels to separate RNA in the range of 500-10000 nt, here in panel E.

All markers and gel types are mentioned in the figure legend. In addition, we have added a new panel 4F to show the significance of change of *cas12k* signal in Northern blot hybridization.

References in this letter

1. Marchler-Bauer, A. & Bryant, S. H. CD-Search: protein domain annotations on the fly. *Nucleic Acids Research* **32**, W327–W331 (2004).
2. Soding, J. Protein homology detection by HMM-HMM comparison. *Bioinformatics* **21**, 951–960 (2005).
3. Hou, S. *et al.* CRISPR-Cas systems in multicellular cyanobacteria. *RNA Biol* **16**, 518–529 (2019).
4. Santamaría-Gómez, J. *et al.* Role of a cryptic tRNA gene operon in survival under translational stress. *Nucleic Acids Research* **49**, 8757–8776 (2021).
5. Athukoralage, J. S. *et al.* The dynamic interplay of host and viral enzymes in type III CRISPR-mediated cyclic nucleotide signalling. *eLife* **9**, e55852 (2020).
6. Millman, A., Melamed, S., Amitai, G. & Sorek, R. Diversity and classification of cyclic-oligonucleotide-based anti-phage signalling systems. *Nat Microbiol* **5**, 1608–1615 (2020).
7. Klucar, L., Stano, M. & Hajduk, M. phiSITE: database of gene regulation in bacteriophages. *Nucleic Acids Research* **38**, D366–D370 (2010).
8. Stano, M. & Klucar, L. phiGENOME: An integrative navigation throughout bacteriophage genomes. *Genomics* **98**, 376–380 (2011).
9. Couñago, R. M. *et al.* Structural basis of thiol-based regulation of formaldehyde detoxification in *H. influenzae* by a MerR regulator with no sensor region. *Nucleic Acids Res* **44**, 6981–6993 (2016).
10. Philips, S. J., Canalizo-Hernandez, M., Yildirim, I., Schatz, G. C. & Mondragón, A. Allosteric transcriptional regulation via changes in the overall topology of the core promoter. *Science* **349**, 877–881 (2015).

11. Krissinel, E. & Henrick, K. Inference of macromolecular assemblies from crystalline state. *J Mol Biol* **372**, 774–797 (2007).
12. Leiva, L. E. & Katz, A. Regulation of leaderless mRNA translation in bacteria. *Microorganisms* **10**, 723 (2022).
13. Moll, I., Grill, S., Gualerzi, C. O. & Bläsi, U. Leaderless mRNAs in bacteria: surprises in ribosomal recruitment and translational control. *Mol Microbiol* **43**, 239–246 (2002).
14. Zheng, X., Hu, G.-Q., She, Z.-S. & Zhu, H. Leaderless genes in bacteria: clue to the evolution of translation initiation mechanisms in prokaryotes. *BMC Genomics* **12**, 361 (2011).
15. Kopf, M. *et al.* Comparative analysis of the primary transcriptome of *Synechocystis* sp. PCC 6803. *DNA Res.* **21**, 527–539 (2014).
16. Mendoza-Vargas, A. *et al.* Genome-wide identification of transcription start sites, promoters and transcription factor binding sites in *E. coli*. *PLoS One* **4**, e7526 (2009).
17. Plate, L. & Marletta, M. A. Phosphorylation-dependent derepression by the response regulator HnoC in the *Shewanella oneidensis* nitric oxide signaling network. *Proceedings of the National Academy of Sciences* **110**, (2013).
18. Travis, B. A. *et al.* Molecular dissection of the glutamine synthetase-GlnR nitrogen regulatory circuitry in Gram-positive bacteria. *Nat Commun* **13**, 3793 (2022).
19. Zhou, W. *et al.* A feedback regulatory loop containing McdR and WhiB2 controls cell division and DNA repair in Mycobacteria. **13**, 14 (2022).
20. Bush, M. J., Chandra, G., Al-Bassam, M. M., Findlay, K. C. & Buttner, M. J. BldC delays entry into development to produce a sustained period of vegetative growth in *Streptomyces venezuelae*. *mBio* **10**, e02812-18 (2019).
21. Godsey, M. H., Baranova, N. N., Neyfakh, A. A. & Brennan, R. G. Crystal structure of MtaN, a global multidrug transporter gene activator. *J Biol Chem* **276**, 47178–47184 (2001).
22. Schumacher, M. A., Chinnam, N. B., Cuthbert, B., Tonthat, N. K. & Whitfill, T. Structures of regulatory machinery reveal novel molecular mechanisms controlling *B. subtilis* nitrogen homeostasis. *Genes & Development* **29**, 451–464 (2015).

REVIEWERS' COMMENTS

Reviewer #1 (Remarks to the Author):

In the revised version of manuscript, I now find a satisfactory description and discussion of bioinformatic methods and results. In general, the manuscript improved significantly, and several additional experiments have been done to stronger justify the conclusions.

I have just a few minor suggestions:

1. Decide where to describe phylogenetic analysis performed for Cas12k and CvkR. Now CvkR is described in both Methods and Figure 2 legend, while Cas12k only in the legend. It is better to describe both in the Methods.
2. Line 176 "small Arc repressors" modify as follows: "small Arc-like repressors that belong to ribbon-helix-helix DNA-binding protein superfamily". And include a reference to the Arc protein and the RHH fold description.
3. Include alignments of Arc_1 and Acr_2 to new Datasets.

Reviewer #2 (Remarks to the Author):

The whole quality of the manuscript has been largely improved. Many new supportive data have been supplemented in the revised manuscript, including additional bioinformatic, genetic, biochemical, and cell-free analyses. Most of my concerns have been addressed. The main concern is the lack of the structure of protein-DNA promoter complex or the one with effector bound, which limits direct demonstration of mechanistic details, however, the authors did show some light on the interactions between CvrK and DNA promoter by comparing with the previous structure and performing mutagenesis studies. The question of if it is qualified for publication in NC needs to be determined by the editors. Overall, the authors did a good job in response to the comments. One additional comment on the manuscript:

The novel dimerization interface is one of the main structural observations in this study, therefore Figure R4 needs to be included in the revised manuscript to support this new proposed dimerization.

Reviewer #3 (Remarks to the Author):

The new proposed CvkR motif has a reasonable length and is fully supported by the provided additional experimental evidence. The revised manuscript has been much improved and fully addressed my concerns.

Two typos in the revised manuscript.

1. Line 350, should 'the promoter regions of CvkR' be 'the promoter region of cas12k' ?
2. Line 399, 'CvkR might one of those MerR-like', a 'be' is missing after might.

Reviewer #4 (Remarks to the Author):

This revised manuscript by Ziemann, *et al.* and Hess is markedly improved and has addressed my previous critique appropriately. The data provide strong evidence for a new family of MerR transcription regulators. There remain a few minor issues that the authors should address.

Figure 7C: The authors should state in the figure legend that the bottom part of this panel is below the top part and next to the right of panel F. As it looks now, panel 7F looks like it has two figure, a left one and a right one. As it is now shown and described in the legend, this is a bit confusing.

Figure 7G: This is not a stereoview. Perhaps the right side was cropped? Please fix or rewrite the figure legend.

Page 17, line 429: "are also possible to participate in DNA..." would be better written as "are also possibly participating in DNA..."

Page 21, line 517: A bit of a typo? "Unlike BldC binds DNA...". Do you mean "Unlike CvkR, BldC binds DNA..."

Page 52, line 1157: "in the absence of presence..." should be "in the absence or presence..."

Point-to-point replies to the reviews of our manuscript NCOMMS-22-21094-A

Reviewer #1 (Remarks to the Author):

In the revised version of manuscript, I now find a satisfactory description and discussion of bioinformatic methods and results. In general, the manuscript improved significantly, and several additional experiments have been done to stronger justify the conclusions.

I have just a few minor suggestions:

1. Decide where to describe phylogenetic analysis performed for Cas12k and CvkR. Now CvkR is described in both Methods and Figure 2 legend, while Cas12k only in the legend. It is better to describe both in the Methods.

Done as suggested, description in the Methods section was adjusted accordingly.

2. Line 176 “small Arc repressors” modify as follows: “small Arc-like repressors that belong to ribbon-helix-helix DNA-binding protein superfamily”. And include a reference to the Arc protein and the RHH fold description.

Rewording done as suggested and two new references inserted (32 and 33).

3. Include alignments of Arc_1 and Acr_2 to new Datasets.

Done as suggested.

Reviewer #2 (Remarks to the Author):

The whole quality of the manuscript has been largely improved. Many new supportive data have been supplemented in the revised manuscript, including additional bioinformatic, genetic, biochemical, and cell-free analyses. Most of my concerns have been addressed. The main concern is the lack of the structure of protein-DNA promoter complex or the one with effector bound, which limits direct demonstration of mechanistic details, however, the authors did show some light on the interactions between CvrK and DNA promoter by comparing with the previous structure and performing mutagenesis studies. The question of if it is qualified for publication in NC needs to be determined by the editors. Overall, the authors did a good job in response to the comments. One additional comment on the manuscript:

The novel dimerization interface is one of the main structural observations in this study, therefore Figure R4 needs to be included in the revised manuscript to support this new proposed dimerization.

Previous Figure R4 is now included in the revised manuscript as Supplementary Fig. 7.

Reviewer #3 (Remarks to the Author):

The new proposed CvkR motif has a reasonable length and is fully supported by the provided additional experimental evidence. The revised manuscript has been much improved and fully addressed my concerns.

Two typos in the revised manuscript.

1. Line 350, should 'the promoter regions of CvkR' be 'the promoter region of cas12k' ?

Yes. Corrected as suggested.

2. Line 399, 'CvkR might one of those MerR-like', a 'be' is missing after might.

Done as suggested.

Reviewer #4 (Remarks to the Author):

This revised manuscript by Ziemann, *et al.* and Hess is markedly improved and has addressed my previous critique appropriately. The data provide strong evidence for a new family of MerR transcription regulators. There remain a few minor issues that the authors should address.

Figure 7C: The authors should state in the figure legend that the bottom part of this panel is below the top part and next to the right of panel F. As it looks now, panel 7F looks like it has two figure, a left one and a right one. As it is now shown and described in the legend, this is a bit confusing.

Many thanks for pointing out this problem. To avoid confusion, we adjusted Fig. 7c and put one panel on the left and the other panel on the right. Then the new Fig. 7 was rearranged with Fig. 7a-c on the top and Fig. 7d-g on the bottom.

Figure 7G: This is not a stereoview. Perhaps the right side was cropped? Please fix or rewrite the figure legend.

Many thanks. We changed it to "Close-up view" in the revised version of our manuscript.

Page 17, line 429: "are also possible to participate in DNA..." would be better written as "are also possibly participating in DNA..."

Done as suggested.

Page 21, line 517: A bit of a typo? "Unlike BldC binds DNA...". Do you mean "Unlike CvkR, BldC binds DNA..."

Yes, this was an instance of awkward wording. Corrected to: "*BldC binds DNA direct repeats as cooperative multimers to regulate gene expression, while both HnoC and McdR are instances...*"

Page 52, line 1157: "in the absence of presence..." should be "in the absence or presence..."

Done as suggested.